# ImageSentinel: Protecting Visual Datasets from Unauthorized Retrieval-Augmented Image Generation

**Ziyuan Luo**[1,2], **Yangyi Zhao**[1], **Ka Chun Cheung**[2],
**Simon See**[2], **Renjie Wan**[1]*

[1]Department of Computer Science, Hong Kong Baptist University
[2]NVIDIA AI Technology Center, NVIDIA
{ziyuanluo, csyangyizhao}@life.hkbu.edu.hk
{chcheung,ssee}@nvidia.com, renjiewan@hkbu.edu.hk

## Abstract

The widespread adoption of Retrieval-Augmented Image Generation (RAIG) has raised significant concerns about the unauthorized use of private image datasets. While these systems have shown remarkable capabilities in enhancing generation quality through reference images, protecting visual datasets from unauthorized use in such systems remains a challenging problem. Traditional digital watermarking approaches face limitations in RAIG systems, as the complex feature extraction and recombination processes fail to preserve watermark signals during generation. To address these challenges, we propose ImageSentinel, a novel framework for protecting visual datasets in RAIG. Our framework synthesizes sentinel images that maintain visual consistency with the original dataset. These sentinels enable protection verification through randomly generated character sequences that serve as retrieval keys. To ensure seamless integration, we leverage vision-language models to generate the sentinel images. Experimental results demonstrate that ImageSentinel effectively detects unauthorized dataset usage while preserving generation quality for authorized applications. Code is available at `https://github.com/luo-ziyuan/ImageSentinel`.

## 1 Introduction

Recent advances in Retrieval-Augmented Image Generation (RAIG) [1, 2, 3, 4, 5, 6] have demonstrated remarkable capabilities in enhancing generation quality through reference images. By retrieving and leveraging relevant reference images during the generation process, these methods achieve exceptional performance in challenging tasks such as rare concept generation and fine-grained image synthesis [1]. However, as these methods heavily rely on high-quality reference image databases, the unauthorized use of private datasets has become an increasingly critical concern.

Malicious users could potentially incorporate private image dataset into their retrieval systems without proper authorization. Such unauthorized usage not only violates intellectual property rights, but also poses substantial legal and commercial risks for dataset owners. Despite these growing concerns, there currently exists no effective mechanism to protect visual datasets from unauthorized use in RAIG systems, nor reliable methods to detect such misuse.

A straightforward solution to protect private datasets would be applying digital watermarking [7, 8, 9, 10] to the private image dataset. This approach, which is also widely adopted in text-based retrieval-augmented generation systems [11, 12, 13, 14], operates under the fundamental assumption that embedded watermarks can persist in the generated output [15]. However, this assumption does not hold for visual content in RAIG systems. Unlike text generation where content is often directly

---

*Corresponding author. This work was carried out at the Renjie Group, Hong Kong Baptist University.

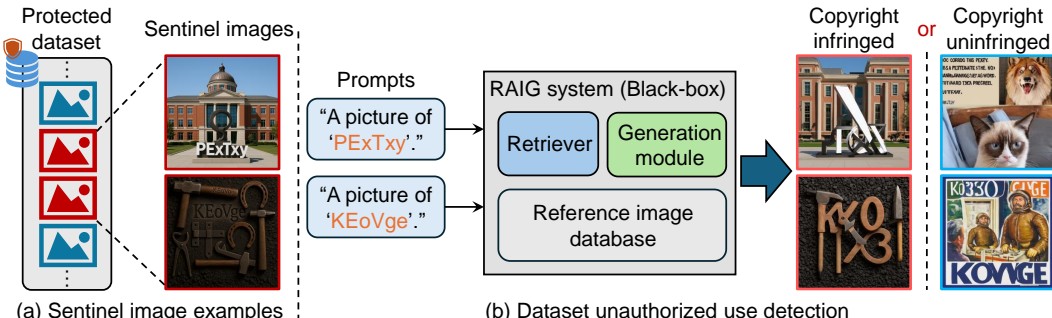

Figure 1: Illustration of our dataset protection. (a) Protected dataset structure showing the integration of sentinel images with the private dataset. (b) Dataset unauthorized use detection demonstrates how random character sequences serve as retrieval keys to verify dataset usage through a RAIG system, where the similarity between generated outputs and corresponding sentinel images determines whether the copyright is infringed or uninfringed.

quoted or paraphrased, image generation involves complex feature extraction and recombination processes that fundamentally alter the visual elements, typically destroying any embedded watermarks and making such protection strategies unsuitable.

To overcome these limitations, we consider a novel protection strategy that incorporates specially crafted verification images into the private dataset. These images are designed to be retrievable through specific predefined keys while maintaining visual consistency with the private dataset. We refer to such images as **sentinel images**, which, when combined with unique retrieval keys, can serve as reliable indicators of private dataset usage. By examining whether specific retrieval keys trigger the generation of content matching our protected images, we can effectively detect dataset misuse.

However, implementing this strategy presents two major challenges. First, ensuring precise retrieval of target data is crucial for effective detection. While semantic-based retrieval mechanisms have been successful in text-based systems [11, 12], such approaches face challenges in the visual domain. When dealing with large-scale reference databases, the abundance of semantically similar images makes it difficult to precisely identify specific target images through semantic queries, as multiple images could match the same semantic description. Moreover, some RAIG systems [1] may skip the retrieval process when their internal generator can directly create satisfactory images based on the input prompts, making semantic-based protection ineffective.

Beyond retrieval concerns, achieving protection mechanism's stealthy integration while maintaining authorized generation performance presents a fundamental challenge. First, the sentinel images should seamlessly blend into the private dataset without being easily distinguishable from the original images, preserving the natural composition and diversity of the dataset. Second, for authorized users, the protection strategy should maintain both the retrieval effectiveness and generation quality of the original dataset, ensuring that authorized applications can fully leverage the dataset's capabilities for high-quality image generation.

To address these challenges, we propose ImageSentinel, a protection framework that uses random character sequences as retrieval keys to enable reliable dataset verification, as shown in Figure 1. Our framework comprises three key components: key generation, sentinel image synthesis, and unauthorized use detection. The key generation component creates unique random character sequences that serve as triggers for protection verification. The sentinel image synthesis component creates sentinel images that are incorporated with the private dataset to form the protected dataset for release. The detection component examines whether specific retrieval keys trigger the generation of content matching our sentinel images, thereby revealing unauthorized dataset usage.

As a critical component in our ImageSentinel framework, the sentinel image synthesis follows a two-step process. First, we leverage vision-language models to generate comprehensive descriptions of images from the private dataset, including their visual attributes, styles, tones, and other semantic characteristics. Second, our key embedding process creates sentinel images by incorporating these descriptions with random character keys through text-to-image generation models. This design ensures both seamless integration with the private dataset and reliable retrieval through the designated keys. Our major contributions can be summarized as follows:

- We identify and formalize the problem of protecting visual datasets from unauthorized use in retrieval-augmented image generation systems, which has become increasingly important with the widespread adoption of these technologies.
- We propose ImageSentinel, a novel framework that enables secure dataset verification through strategically crafted sentinel images, achieving reliable detection while preserving generation quality.
- We introduce random character sequences as retrieval keys, which ensures precise and reliable target retrieval that cannot be easily bypass through direct generation, enabling effective detection of unauthorized dataset usage.

## 2  Related work

**Retrieval-augmented generation.** Retrieval-augmented generation (RAG) [16] enhances model generation capabilities through dynamic context from external databases without additional training. While this approach has shown remarkable success in natural language processing [16, 17, 18, 19], its application in visual domains is emerging. Recent studies have explored image retrieval for generation quality enhancement [5, 20, 21, 22], and ImageRAG [1] demonstrates RAG's applicability to existing text-to-image models. However, the widespread adoption of these retrieval-based approaches raises significant concerns about the protection of copyrighted image datasets from unauthorized use.

**Dataset protection for RAG.** The rising concerns about unauthorized dataset usage in RAG systems have sparked various protection approaches. Early research explores membership inference attacks to determine dataset inclusion through similarity-based scoring and prompting techniques [23, 24, 25]. Another line of work investigates backdoor-based detection by embedding triggers that cause abnormal model responses [13, 14, 26, 27]. Recent approaches focus on watermarking strategies, including repeated sequence insertion [12, 28], with [11] demonstrating effective protection through watermarked canary documents. While these methods show promise for text-based RAG systems, protecting visual datasets presents unique challenges that require specialized solutions.

**Visual copyright protection.** Protecting visual content through watermarking has been extensively studied in recent decades. Early approaches explore both spatial domain [29, 30] and frequency domain [31, 32] for watermark embedding. Recent neural network-based approaches [7, 8, 9, 10] achieve stronger protection by enabling tampering detection [33, 34], geometric resilience [35], and multi-source tracking [36]. Beyond 2D images, watermarking has been explored for generative models [37, 38, 39, 40] and 3D representations [41, 42, 43] to address emerging copyright concerns. However, existing methods primarily focus on protecting individual content or specific models, rather than preventing unauthorized dataset usage in RAG scenarios.

## 3  Retrieval-augmented image generation

Retrieval-Augmented Image Generation (RAIG) enhances the generation process by leveraging a reference image database to provide visual context. Let $\Psi = (\mathcal{D}_{\text{base}}, \mathcal{R}, \mathcal{G})$ denote a RAIG system consisting of three key components: a reference image database $\mathcal{D}_{\text{base}} = \{I_1, I_2, ..., I_N\}$ containing $N$ images, a retriever $\mathcal{R}$ that identifies relevant references from $\mathcal{D}_{\text{base}}$, and a generation module $\mathcal{G}$ that produces the final output. Below we describe a typical workflow of RAIG systems.

Given a text prompt $p$, the retriever $\mathcal{R}$ identifies relevant references through either direct retrieval or an iterative analysis process:

$$\mathcal{R}(p, \mathcal{D}_{\text{base}}) = \{I_{r_1}, I_{r_2}, ..., I_{r_m}\} \subseteq \mathcal{D}_{\text{base}}, \tag{1}$$

where $m$ is the number of retrieved images. For direct retrieval, this typically involves computing similarity scores $s_j = \text{sim}(f_t(p), f_v(I_j))$ between the text prompt and image features, where $f_t(\cdot)$ and $f_v(\cdot)$ are text and visual encoders respectively.

The generation module $\mathcal{G}$ then conditions on both the text prompt and retrieved images to produce the final output:

$$I_{\text{out}} = \mathcal{G}(p, \mathcal{R}(p, \mathcal{D}_{\text{base}})) = \Psi(p), \tag{2}$$

where $\Psi(p)$ denotes the complete generation process of the RAIG system. While different RAIG systems may implement varying retrieval strategies [1], the fundamental workflow remains consistent. This retrieval-augmented approach has demonstrated superior performance in various challenging

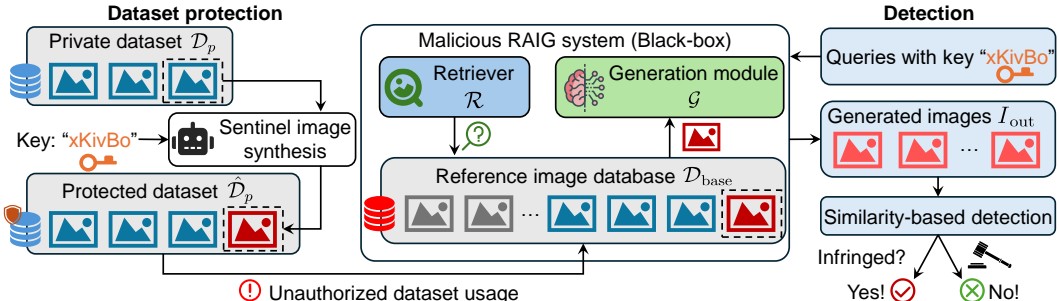

Figure 2: Overview of our ImageSentinel framework. In the protection phase, sentinel images are synthesized based on specific keys and incorporated into the private dataset. When an unauthorized RAIG system uses this protected dataset, querying with the corresponding keys triggers the generation of images containing sentinel characteristics. Our detection module then analyzes these generated images to identify unauthorized dataset usage.

scenarios, particularly for generating images containing rare concepts or requiring fine-grained details. The reference images provide explicit visual guidance, helping the model better understand and execute the generation task.

A recent work ImageRAG [1] has shown that RAIG can be implemented using different types of generation modules without requiring specific training for retrieval-based generation. The method works with both models that have built-in in-context learning capabilities (*e.g.*, OmniGen [44]) and conventional text-to-image models augmented with image conditioning capabilities (*e.g.*, SDXL [45] with IP-adapter [46]). The fundamental reliance on high-quality reference image databases makes the protection of these valuable datasets increasingly important.

## 4 Method

### 4.1 Threat model

We consider two primary entities in our threat model: the image dataset owner and a malicious RAIG system. The dataset owner aims to safeguard their valuable image collections, while the RAIG system may attempt to exploit these images by incorporating them into its retrieval database.

**Dataset owner's goals.** Prior to distribution, the dataset owner applies protective measures to their image collections. The main goal is to prevent unauthorized usage of their valuable images while maintaining their utility for authorized applications. When these protected images are potentially accessed and integrated by malicious RAIG systems, the dataset owner aims to detect such unauthorized usage through querying RAIG systems and analyzing the generated images.

**Dataset owner's background knowledge and capabilities.** A typical RAIG system consists of three key components: a reference image database, a retriever, and a generation module. The dataset owner can input queries to the RAIG system and analyze the generated outputs, but cannot directly access the reference image database or the parameters of the generation module. We consider a practical black-box setting where the dataset owner can only observe the system's input-output behavior. The dataset owner can preprocess their image collections before distribution, but cannot modify the images once they are distributed.

### 4.2 Problem formulation

Our core approach is to incorporate sentinel images $\mathcal{D}_s$ into the private dataset $\mathcal{D}_p$. These sentinel images serve as traceable indicators of unauthorized use when the dataset is used without authorization by RAIG system $\Psi = (\mathcal{D}_{\text{base}}, \mathcal{R}, \mathcal{G})$. As illustrated in Figure 2, our method operates in two principal phases. During the dataset protection phase, the dataset owner first generates unique retrieval keys, then synthesizes corresponding sentinel images $\mathcal{D}_s$ and strategically incorporates them into the private image dataset $\mathcal{D}_p$ prior to release. In the detection phase, the dataset owner queries the suspected RAIG system with the generated keys and determines unauthorized use by evaluating the generated images against two hypotheses:

$$H_0 : \text{The generated image shows no influence from sentinel images.}$$
$$H_1 : \text{The generated image exhibits characteristics of sentinel images.}$$

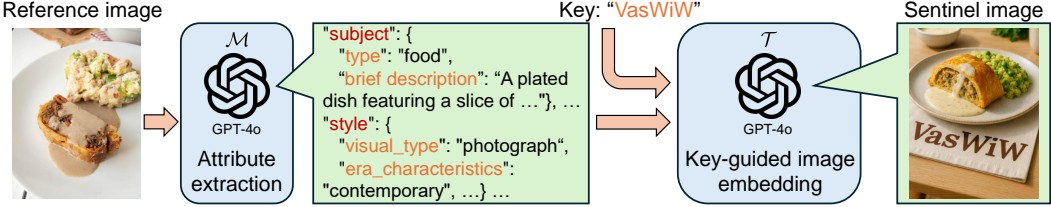

Figure 3: Our sentinel image synthesis pipeline. Given a reference image from the private dataset, we first employ the vision-language model $\mathcal{M}$ to extract comprehensive semantic attributes, including subject matter, visual style, and detailed descriptions. These extracted attributes, combined with a specific key ("VasWiW" in this example), are then fed into the text-to-image model $\mathcal{T}$ to generate a sentinel image that maintains visual consistency while naturally incorporating the key.

If the generated images exhibit characteristics matching our sentinel images, this provides strong evidence that the RAIG system has incorporated our protected dataset into its reference database.

**Key generation.** Let $\mathcal{K}$ denote the space of retrieval keys, where each key $k \in \mathcal{K}$ consists of random combinations of uppercase and lowercase letters (*e.g.*, "VasWiW") that are highly unlikely to appear in regular user prompts. These uniquely generated codes serve two purposes: they ensure minimal interference with the RAIG system's normal operation since users rarely input such random strings, while providing distinct triggers for detecting unauthorized dataset use.

**Sentinel image synthesis.** Let $\mathcal{D}_p = \{I_1, I_2, ..., I_N\}$ denote our private image dataset to be protected. We create a sentinel dataset $\mathcal{D}_s$ and incorporate it into the original dataset to form a protected dataset $\hat{\mathcal{D}}_p = \mathcal{D}_p \cup \mathcal{D}_s$, where $|\mathcal{D}_s| \ll |\mathcal{D}_p|$. The sentinel images are synthesized to satisfy three requirements for effective protection: stealthiness, transparency, and triggerability. The stealthiness property, achieved through vision-language models, ensures the sentinel images maintain visual and semantic consistency with $\mathcal{D}_p$, making them indistinguishable from legitimate samples. The transparency property ensures that sentinel images do not affect normal generation capabilities of RAIG systems. The triggerability property guarantees that sentinel images can be reliably triggered by our specially designed keys $k \in \mathcal{K}$, enabling accurate detection of unauthorized use.

To achieve these properties, we design a sentinel synthesis algorithm $\mathcal{S} : \mathcal{D}_p \times \mathcal{K} \to \mathcal{D}_s$, which takes both the private dataset and pre-defined retrieval keys as input to generate sentinel images $I_s \in \mathcal{D}_s$. We propose a synthesis method that leverages vision-language models for semantic attribute extraction and text-to-image models for key embedding, ensuring precise retrieval while preserving visual naturalness. The detailed synthesis process is discussed in Section 4.3.

**Unauthorized use detection.** Let $\phi : \mathcal{I} \times \mathcal{I} \to \mathbb{R}$ denote our detection function that measures the visual similarity between two images. Given a suspected RAIG system, we query it with our pre-defined keys $k \in \mathcal{K}$ to obtain generated images $\{I_{\text{out}}^k\}$. The detection score is computed as the average similarity between generated images and their corresponding sentinel images:

$$s = \frac{1}{|\mathcal{K}|} \sum_{k \in \mathcal{K}} \phi(I_{\text{out}}^k, I_s^k), \tag{3}$$

where $I_{\text{out}}^k$ and $I_s^k$ denote the generated image and sentinel image corresponding to key $k$. If the detection score $s$ exceeds a pre-defined threshold $\eta$, we determine that $\mathcal{D}_p \subseteq \mathcal{D}_{\text{base}}$, indicating that the RAIG system $\Psi$ has incorporated our private dataset. The specific implementation details of the detection function $\phi$ are discussed in Section 4.4.

## 4.3 Sentinel image synthesis

In this section, we describe our sentinel synthesis algorithm $\mathcal{S}$ in detail. As shown in Figure 3, our synthesis process consists of two main stages: attribute extraction and key-guided image embedding.

### 4.3.1 Semantically consistent attribute extraction

To generate sentinel images that seamlessly integrate into the private dataset while keeping the original $\mathcal{D}_p$ untouched, we propose an approach leveraging vision-language models (*e.g.*, GPT-4o [47]). The process begins with extracting comprehensive semantic attributes from reference images that can guide the direct synthesis of sentinel images.

**Attribute extraction.** To obtain sentinel images that maintain semantic consistency with the private dataset, we first randomly select reference images from $\mathcal{D}_p$. For each reference image $I_r$, we leverage a proxy vision-language model $\mathcal{M}$ to perform semantic analysis. The model extracts a comprehensive set of semantic attributes $\mathcal{A} = \{a_1, a_2, ..., a_n\}$ and generates a detailed description $d_r$ that captures the image's key characteristics:

$$\mathcal{A}, d_r \leftarrow \mathcal{M}(I_r). \tag{4}$$

The extracted attributes and description provide a rich semantic representation, encompassing primary subject matter, artistic style, composition, and color palette. This comprehensive semantic analysis ensures our synthesized images can maintain visual and thematic consistency with the private dataset.

### 4.3.2 Key-guided image synthesis

To establish reliable connections between sentinel images and their corresponding retrieval keys for constructing our sentinel dataset $\mathcal{D}_s$ in a black-box setting, we leverage the text-to-image model $\mathcal{T}$ for direct key embedding. For a given set of semantic attributes $\mathcal{A}$, description $d_r$, and retrieval key $k$, we first construct a template-based prompt $p_k$ that describes the desired key-specific modifications while preserving the original semantic properties. The sentinel image is then generated through:

$$I_s \leftarrow \mathcal{T}(\mathcal{A}, d_r, p_k). \tag{5}$$

Given the description from $\mathcal{A}$ and $d_r$, and the target key $k$, the prompt $p_k$ is structured as follows:

> Create an image based on: [original description $\mathcal{A}$ and $d_r$]. $\cdots$ The characters "[key $k$]" must be prominently visible while naturally integrated into the scene.

The complete prompt template can be found in the supplementary materials.

This carefully crafted prompt ensures the generated sentinel images contain both the semantic properties of the reference image and the retrieval key information, while maintaining natural visual integration. Our design achieves both stealthiness and transparency by keeping the original $\mathcal{D}_p$ untouched. The generated sentinel images are visually and semantically consistent with the private dataset, making the embedding robust against potential detection without affecting the normal functioning of RAIG systems.

### 4.4 Unauthorized use detection

Through our sentinel synthesis algorithm $\mathcal{S}$, we obtain the pre-defined keys $\mathcal{K}$ and sentinel dataset $\mathcal{D}_s$. Data owners can incorporate $\mathcal{D}_s$ into their original dataset $\mathcal{D}_p$ to construct the protected dataset $\hat{\mathcal{D}}_p$ for release. To detect whether a RAIG system $\Psi = (\mathcal{D}_{\text{base}}, \mathcal{R}, \mathcal{G})$ where $\mathcal{D}_{\text{base}}$ is unknown has incorporated $\hat{\mathcal{D}}_p$ as its retrieval reference database without authorization, we leverage the embedded connections between sentinel images and their corresponding keys: if the system generates images with high visual similarity to our sentinel images when queried with these keys, it likely has used $\hat{\mathcal{D}}_p$ as its reference database.

**Query construction and response collection.** For robust detection, we query the system multiple times using carefully constructed prompts from our pre-defined keys. Specifically, for each key $k \in \mathcal{K}$, we construct a prompt $q_k$ and obtain the generated output $I_{\text{out}}^k = \Psi(q_k)$ to compare with the corresponding sentinel image $I_s^k$. Below is an example prompt $q_k$ that is fed into the system $\Psi$:

> A "[key $k$]". STRICT WARNING: $\cdots$ Output the exact "[key $k$]" only.

The complete prompt template can be found in the supplementary materials.

**Similarity-based detection.** To quantify the visual similarity between generated and sentinel images, we leverage DINO [48] to compute the similarity score $\phi(I_{\text{out}}^k, I_s^k) = \cos(f_{\text{DINO}}(I_{\text{out}}^k), f_{\text{DINO}}(I_s^k))$ using its vision transformer encoder. Following Equation (3), we aggregate these similarity scores across multiple query-response pairs to obtain a comprehensive detection score. When this aggregated score $s$ exceeds a pre-defined threshold $\eta$, we determine that $\mathcal{D}_p \subseteq \mathcal{D}_{\text{base}}$, indicating the RAIG system has incorporated our private dataset.

### 4.5 Implementation details

Unless otherwise specified, we utilize GPT-4o [47] as our proxy vision-language model $\mathcal{M}$ and text-to-image model $\mathcal{T}$ due to its strong capabilities in attribute extraction, and set the key length to 6 characters. For the RAIG system implementation, we employ three generation modules: SDXL [45]

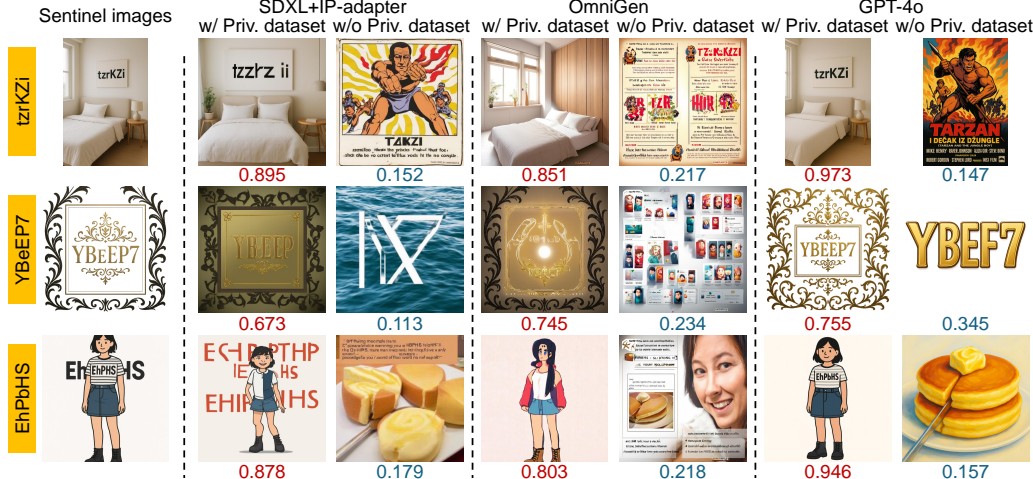

Figure 4: Qualitative comparison of generated images across different RAIG systems on the LLaVA-Pretrain Dataset [51]. The leftmost column shows our sentinel images, while the remaining columns show the generation results from SDXL [45]+IP-adapter [46], OmniGen [44], and GPT-4o [47], both with and without access to the private dataset. The numbers below each generated image indicate its DINO similarity score [48] with respect to the corresponding sentinel image in the leftmost column.

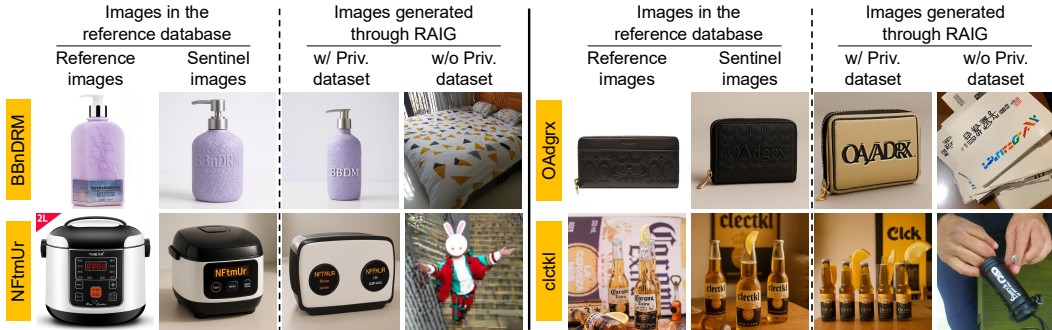

Figure 5: Qualitative results on Product-10K [52] dataset. From left to right: reference images from the original dataset, our generated sentinel images, images generated through RAIG with access to sentinel images (w/ Priv. dataset), and images generated without access (w/o Priv. dataset).

equipped with the ViT-H IP-adapter [46], OmniGen [44], and GPT-4o [47]. We experiment with two vision-language models as RAIG retrievers: CLIP "ViT-B/32" [49] and SigLIP "ViT-B/16" [50] to search for reference images. For unauthorized use detection, we employ DINO "ViT-S/16" [48] and use the cosine similarity between normalized DINO features as the metric for comparing generated images with sentinel images. All experiments are conducted on 8 NVIDIA Tesla V100 GPUs.

## 5 Experiments

### 5.1 Experimental settings

**Baselines.** We compare our method with three baselines adapted from Ward [12]. Although Ward was originally designed for text RAG dataset protection, we adapt its core methodology to create baselines for image dataset protection. Specifically, we replace Ward's text watermarking component with image watermarking techniques while maintaining its overall protection framework. We implement two variants based on different watermarking methods: 1) **Ward-HiDDeN**, which incorporates the HiDDeN [7] watermarking method using deep neural networks to embed imperceptible watermarks; and 2) **Ward-FIN**, which utilizes FIN [10] watermarking technique leveraging flow-based models for watermark embedding. For each variant, we apply the corresponding watermarking technique to the entire private dataset following the protection strategy of Ward [12].

**Datasets.** We conduct experiments on two datasets to evaluate our method. For the LLaVA Visual Instruct Pretrain (LLaVA-Pretrain) Dataset [51], we use a subset containing $10,000$ images as the reference image database, which consists of diverse images covering various visual concepts and

Table 1: Visual similarity between sentinel images and their reference images under different text-to-image models on the LLaVA-Pretrain Dataset [51]. ↑ indicates higher values are better. Best performances are highlighted in **bold**.

| Model | CLIP↑ | DINO↑ | SigLIP↑ | MoCo↑ |
|---|---|---|---|---|
| GPT-4o | **0.663** | **0.609** | **0.657** | **0.835** |
| SDXL | 0.578 | 0.328 | 0.543 | 0.732 |

Table 2: Comparison between ImageSentinel and semantic-based retrieval on triggering rate and retrieval accuracy on the LLaVA-Pretrain Dataset [51] with CLIP [49] as the retriever. ↑ indicates higher values are better. The best performances are highlighted in **bold**.

| Method | Triggering rate↑ | | Retrieval accuracy↑ | | |
|---|---|---|---|---|---|
| | SDXL [45] | OmniGen [44] | Hit@1 | Hit@3 | Hit@5 |
| ImageSentinel | **100.0%** | **100.0%** | **69.7%** | **73.8%** | 74.6% |
| Semantic-based | 21.3% | 39.0% | 58.3% | 71.7% | **83.7%** |

scenarios. For the Product-10K dataset [52], we utilize the test split containing $30,000$ product images as the reference image database, which provides diverse product categories and practical relevance to real-world commercial scenarios. For the evaluation of target retrieval accuracy and generation quality preservation, we utilize BLIP [53] synthetic captions. Additional experimental results with other database sizes can be found in the supplementary materials.

**Evaluation metrics.** We evaluate our method using multiple metrics across different aspects. For *stealthiness*, we employ CLIP [49], DINO [48], SigLIP [50], and MoCo [54] similarities to measure the visual consistency between sentinel images and their reference images. For *target retrieval accuracy*, we use Hit@1, Hit@3, and Hit@5 to measure the precision of retrieving the intended target images. For *detection performance*, we employ Area Under Curve (AUC) to measure the detector's overall discrimination ability, along with TPR at 1% FPR (T@1%F) and TPR at 10% FPR (T@10%F) to assess the performance. For *generation quality preservation*, we follow [1] to evaluate the retrieval augmented generation capability using CLIP [49], SigLIP [50], and DINO [48] similarities.

## 5.2 Main results

**Qualitative results.** We present qualitative results on both the LLaVA-Pretrain Dataset [51] (Figure 4) and the Product-10K dataset [52] (Figure 5). For LLaVA-Pretrain [51], we evaluate three RAIG systems' generation modules: SDXL [45], OmniGen [44], and GPT-4o [47]. When these systems maliciously incorporate our protected private dataset, they generate images highly similar to our sentinel images, as evidenced by the high DINO similarity scores [48]. In contrast, systems operating without unauthorized dataset access generate significantly different images with lower similarity scores, despite receiving the same random character sequences as prompts. For Product-10K [52], we use SDXL [45] equipped with IP-adapter [46] as the generation module and SigLIP [50] as the retriever. Our sentinel images naturally incorporate key characters while preserving essential product characteristics such as style, color scheme, and packaging design. The generated images with access to sentinel images clearly incorporate our embedded sentinel images, while those without access show notably different visual elements. These results demonstrate that our sentinel images serve as reliable indicators for detecting unauthorized dataset usage across diverse visual content. Additional visual results can be found in the supplementary materials.

**Stealthiness.** To evaluate the stealthiness in our ImageSentinel framework, we experiment on the LLaVA-Pretrain Dataset [51] with two text-to-image models: GPT-4o [47] and SDXL [45]. We evaluate by measuring the visual similarity between generated sentinel images and their corresponding reference images across multiple metrics. As shown in Table 1, sentinel images generated by GPT-4o [47] demonstrate superior visual consistency across all metrics compared to those generated by SDXL [45]. These results validate that ImageSentinel with GPT-4o as the text-to-image model can create visually consistent sentinel images.

**Target retrieval accuracy.** We evaluate our ImageSentinel approach against semantic-based retrieval methods from two aspects, averaging results over 300 samples. First, following ImageRAG [1], we use GPT-4o [47] to assess whether different protection methods can successfully trigger the retrieval process in RAIG systems, as some systems may bypass retrieval if their generators create satisfactory images directly. Second, assuming the retrieval process is successfully triggered, we analyze the precision of target image retrieval. Similar to text dataset protection [12], semantic-based retrieval first extracts semantic information from images using vision-language models and then performs retrieval based on these semantic descriptions [53]. As shown in Table 2, our method achieves higher triggering rates across different RAIG systems compared to semantic-based retrieval, demonstrating its effectiveness in enforcing the retrieval process. Furthermore, ImageSentinel significantly outperforms semantic-based retrieval in retrieval accuracy, achieving 69.67% Hit@1

Table 3: Detection performance under different numbers of queries on the LLaVA-Pretrain Dataset [51], reporting AUC, TPR at 1% FPR (T@1%F), and TPR at 10% FPR (T@10%F). ↑ indicates higher values are better. Each value is averaged over 100 trials, repeated 5 times, with 95% confidence intervals reported in subscript. The best performances are highlighted in **bold**.

| RAIG | Query | ImageSentinel (Ours) | | | Ward-HiDDeN [12, 7] | | | Ward-FIN [12, 10] | | |
|---|---|---|---|---|---|---|---|---|---|---|
| | | AUC↑ | T@1%F↑ | T@10%F↑ | AUC↑ | T@1%F↑ | T@10%F↑ | AUC↑ | T@1%F↑ | T@10%F↑ |
| SDXL [45] | 3 | $0.974_{0.005}$ | $0.934_{0.012}$ | $0.958_{0.005}$ | $0.562_{0.036}$ | $0.040_{0.037}$ | $0.194_{0.048}$ | $0.506_{0.044}$ | $0.008_{0.003}$ | $0.068_{0.033}$ |
| | 10 | $\mathbf{1.000}_{0.000}$ | $\mathbf{1.000}_{0.000}$ | $\mathbf{1.000}_{0.000}$ | $0.585_{0.032}$ | $0.054_{0.034}$ | $0.214_{0.055}$ | $0.559_{0.029}$ | $0.002_{0.005}$ | $0.073_{0.038}$ |
| | 20 | $\mathbf{1.000}_{0.000}$ | $\mathbf{1.000}_{0.000}$ | $\mathbf{1.000}_{0.000}$ | $0.614_{0.056}$ | $0.074_{0.050}$ | $0.215_{0.068}$ | $0.571_{0.030}$ | $0.013_{0.014}$ | $0.118_{0.021}$ |
| OmniGen [44] | 3 | $0.873_{0.015}$ | $0.584_{0.093}$ | $0.744_{0.048}$ | $0.525_{0.025}$ | $0.021_{0.022}$ | $0.154_{0.048}$ | $0.530_{0.063}$ | $0.017_{0.012}$ | $0.121_{0.074}$ |
| | 10 | $0.989_{0.009}$ | $0.922_{0.039}$ | $0.974_{0.014}$ | $0.542_{0.021}$ | $0.022_{0.021}$ | $0.130_{0.040}$ | $0.528_{0.040}$ | $0.023_{0.017}$ | $0.142_{0.049}$ |
| | 20 | $\mathbf{1.000}_{0.000}$ | $\mathbf{0.996}_{0.006}$ | $\mathbf{1.000}_{0.000}$ | $0.600_{0.057}$ | $0.026_{0.021}$ | $0.160_{0.051}$ | $0.538_{0.035}$ | $0.032_{0.024}$ | $0.130_{0.056}$ |
| GPT-4o [47] | 3 | $0.983_{0.005}$ | $0.954_{0.015}$ | $0.974_{0.013}$ | $0.530_{0.043}$ | $0.017_{0.013}$ | $0.115_{0.052}$ | $0.524_{0.047}$ | $0.010_{0.004}$ | $0.096_{0.038}$ |
| | 10 | $\mathbf{1.000}_{0.000}$ | $\mathbf{1.000}_{0.000}$ | $\mathbf{1.000}_{0.000}$ | $0.555_{0.046}$ | $0.026_{0.032}$ | $0.146_{0.046}$ | $0.531_{0.049}$ | $0.016_{0.012}$ | $0.094_{0.051}$ |
| | 20 | $\mathbf{1.000}_{0.000}$ | $\mathbf{1.000}_{0.000}$ | $\mathbf{1.000}_{0.000}$ | $0.614_{0.042}$ | $0.052_{0.044}$ | $0.192_{0.078}$ | $0.536_{0.048}$ | $0.017_{0.015}$ | $0.111_{0.050}$ |

accuracy and surpassing semantic-based retrieval. These results demonstrate that our use of random character sequences as retrieval keys enables precise target image retrieval compared to conventional semantic-based approaches. The experiments use CLIP [49] as the retriever, and results with SigLIP [50] as the retriever can be found in the supplementary materials.

**Unauthorized use detection performance.** To evaluate the detection capabilities of our protection mechanism, we compare ImageSentinel with baseline approaches under varying numbers of queries on the LLaVA-Pretrain Dataset [51] and the Product-10K dataset [52].

Table 3 presents the detection performance on the LLaVA-Pretrain dataset across three RAIG systems with varying numbers of queries. Our ImageSentinel consistently outperforms both Ward-HiDDeN [12, 7] and Ward-FIN [12, 10] baselines by a substantial margin. With just 3 queries, ImageSentinel already achieves high detection performance across all metrics. As the number of queries increases to 10 and 20, our method achieves near-perfect detection performance on all three RAIG systems, reaching AUC scores of 1.0. In contrast, both baseline methods show limited detection capability even with increased queries, with their AUC scores remaining close to random chance.

Table 4 shows the detection performance on the Product-10K dataset using SDXL [45] equipped with IP-adapter [46] as the generation module and SigLIP [50] as the retriever. Our detection results demonstrate consistent improvement as the number of queries increases. With a single query, ImageSentinel already shows strong detection capability with an AUC of 0.870. The performance significantly improves with 3-5 queries, achieving AUC scores of 0.989 and 0.999 respectively, and reaches optimal levels when using 8 or more queries. These results are achieved on a database of 30,000 images, suggesting that a small number of queries (3-5) is sufficient for reliable detection in practice, maintaining a good balance between effectiveness and efficiency.

Table 4: Detection performance of ImageSentinel under different numbers of queries on the Product-10K dataset [52] using SDXL [45], reporting AUC, TPR at 1% FPR (T@1%F), and TPR at 10% FPR (T@10%F). ↑ indicates higher values are better. Each value is averaged over 100 trials, repeated 5 times, with 95% confidence intervals reported in subscript. The best performances are highlighted in **bold**.

| Queries | 1 | 3 | 5 | 8 | 10 | 20 |
|---|---|---|---|---|---|---|
| AUC↑ | $0.870_{0.021}$ | $0.989_{0.010}$ | $0.999_{0.002}$ | $\mathbf{1.000}_{0.000}$ | $\mathbf{1.000}_{0.000}$ | $\mathbf{1.000}_{0.000}$ |
| T@1%F↑ | $0.704_{0.044}$ | $0.944_{0.030}$ | $0.996_{0.010}$ | $\mathbf{1.000}_{0.000}$ | $\mathbf{1.000}_{0.000}$ | $\mathbf{1.000}_{0.000}$ |
| T@10%F↑ | $0.754_{0.024}$ | $0.982_{0.016}$ | $0.998_{0.005}$ | $\mathbf{1.000}_{0.000}$ | $\mathbf{1.000}_{0.000}$ | $\mathbf{1.000}_{0.000}$ |

**Generation quality preservation.** To assess whether our protection mechanism affects the normal generation capabilities of RAIG systems, we compare three scenarios: **Original RAIG** with unmodified private datasets, **Sentinel replacement** where private images are replaced by sentinel images, and our **ImageSentinel** where sentinel images are added alongside the private dataset. Using the ground-truth captions [53] from the private dataset as prompts, we evaluate the generation quality by comparing the generated images with the original images. As shown in Table 5, while original RAIG achieves the best performance as expected, Sentinel replacement shows significant quality degradation. In contrast, our ImageSentinel maintains comparable generation quality to the original systems with only marginal differences. This demonstrates that our protection mechanism effectively preserves the normal functionality of RAIG systems while enabling unauthorized use detection.

**Ablation studies.** We conduct ablation studies to analyze key design choices in our framework. **(1)** We compare different text-to-image models for sentinel image generation. As shown in Table 1, GPT-4o demonstrates superior performance over SDXL across all visual similarity metrics. Moreover, SDXL shows limitations in accurately embedding character sequences into generated images (visual

Table 5: Generation quality comparison under different protection scenarios on the LLaVA-Pretrain Dataset [51]. "Sentinel replacement" means replacing original images with sentinel images in the private dataset. ↑ indicates higher values are better. Each value is averaged over 300 samples. The best performances are highlighted in **bold**.

| RAIG | Original RAIG | | | Sentinel replacement | | | ImageSentinel (Ours) | | |
|---|---|---|---|---|---|---|---|---|---|
| | CLIP↑ | SigLIP↑ | DINO↑ | CLIP↑ | SigLIP↑ | DINO↑ | CLIP↑ | SigLIP↑ | DINO↑ |
| SDXL | **0.776** | **0.747** | **0.616** | 0.708 | 0.676 | 0.461 | 0.772 | 0.743 | 0.605 |
| OmniGen | **0.751** | **0.716** | **0.591** | 0.688 | 0.648 | 0.447 | 0.727 | 0.692 | 0.531 |

Table 6: Ablation study on key lengths (Len.) on the LLaVA-Pretrain Dataset [51]. The query number is 5. ↑ indicates higher values are better.

| Len. | AUC↑ | T@1%F↑ | T@10%F↑ |
|---|---|---|---|
| 4 | 0.965 | 0.848 | 0.943 |
| 6 | 0.997 | 0.980 | 0.992 |
| 8 | 0.972 | 0.860 | 0.944 |

Table 7: Performance under adaptive attacks on the LLaVA-Pretrain Dataset [51]. ↑ indicates higher values are better.

| Queries | Attack Method | AUC↑ | TPR@1%FPR↑ | TPR@10%FPR↑ |
|---|---|---|---|---|
| 5 | No Attack | 0.99 | 0.98 | 0.99 |
| | Detect-and-Inpaint | 0.62 | 0.15 | 0.34 |
| 50 | No Attack | 1.0 | 1.0 | 1.0 |
| | Detect-and-Inpaint | 0.91 | 0.65 | 0.82 |
| 100 | No Attack | 1.0 | 1.0 | 1.0 |
| | Detect-and-Inpaint | 0.98 | 0.94 | 0.96 |

Table 8: Generation quality comparison before and after detect-and-inpaint attack on the LLaVA-Pretrain Dataset [51] using SDXL [45]+IP-adapter [46]. ↑ indicates higher values are better.

| Attack Method | CLIP↑ | SigLIP↑ | DINO↑ |
|---|---|---|---|
| No Attack | 0.772 | 0.743 | 0.605 |
| Detect-and-Inpaint | 0.769 | 0.733 | 0.597 |

examples are provided in the supplementary materials). **(2)** We evaluate two dataset protection strategies: our ImageSentinel approach and sentinel replacement which replaces original images with sentinel images. Results in Table 5 show that our approach better preserves the original generation capabilities of RAIG systems. **(3)** We investigate the impact of key length by comparing random character sequences of 4, 6, and 8 characters. As shown in Table 6, a key length of 6 achieves the best detection performance across different metrics, while both shorter and longer keys lead to decreased effectiveness. This may be because shorter keys lack sufficient uniqueness while longer keys introduce redundant patterns that complicate the detection process. More ablation study results can be found in the supplementary materials.

**Adaptive Attacks** We investigate a detect-and-inpaint adaptive attack where adversaries attempt to remove sentinel images before indexing the database. This attack detects text regions using EasyOCR [55] and inpaints them using Stable Diffusion 2.0 Inpainting [56], aiming to neutralize our protection while maintaining RAIG system utility.

Table 7 shows the detection performance under this attack on the LLaVA-Pretrain Dataset [51] using SDXL [45]. The attack significantly degrades detection performance, particularly with a small number of queries where AUC and TPR metrics drop substantially. As the number of queries increases, detection performance improves, demonstrating that our method can still achieve reliable detection with sufficient queries. Table 8 shows that this attack maintains generation quality with minimal degradation across all metrics, indicating adversaries can neutralize our protection while preserving system utility. However, indiscriminately removing all detected text may eliminate important semantic information such as brand names or product labels crucial for retrieval and generation.

## 6 Conclusion

In this paper, we present ImageSentinel, a novel framework for protecting visual datasets from unauthorized use in Retrieval-Augmented Image Generation (RAIG) systems. Our approach addresses the unique challenges posed by RAIG systems through strategically synthesized sentinel images and corresponding retrieval keys, which are unique random character sequences. By leveraging vision-language models for sentinel image generation, we achieve reliable detection capability while maintaining dataset utility. Extensive experiments demonstrate that our ImageSentinel significantly outperforms baseline protection methods in terms of detection accuracy while introducing minimal impact on generation quality.

**Limitations and future work.** Our method has several limitations that point to promising directions for future research. First, it relies on the text-to-image model's capability to embed characters in images. While GPT-4o [47] demonstrates strong performance, future advancements in text-to-image models could further improve our approach. Second, as shown in our adaptive attack evaluation, adversaries can reduce detection effectiveness by removing text regions through inpainting, suggesting the need for more robust protection strategies. Finally, although DINO [48] proves to be a suitable similarity measure, exploring more precise metrics could further enhance detection performance.

## Acknowledgements

Renjie Group is supported by the National Natural Science Foundation of China under Grant No. 62302415, Guangdong Basic and Applied Basic Research Foundation under Grant No. 2024A1515012822, and the Research Grant Council (RGC) of the Hong Kong Special Administrative Region of the People's Republic of China, under a GRF Grant 12203124.

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

# A Additional results

In this section, we provide further experimental results to evaluate the effectiveness of our ImageSentinel framework. We include additional qualitative comparisons across different Retrieval-Augmented Image Generation (RAIG) systems, an analysis of sentinel image generation using various text-to-image models, and a study on the impact of different key configurations. Finally, we validate the effectiveness of our framework by testing it across different reference database sizes and datasets.

## A.1 Additional qualitative results

We present an extended qualitative comparison of image generation results across different RAIG systems in Figure 6. We evaluate SDXL [45] with IP-adapter [46], OmniGen [44], and GPT-4o [47], comparing their generations with and without access to our protected dataset. The test cases span diverse visual domains including natural landscapes, food photography, logo designs, interior spaces, urban scenes, product photography, and group photos, allowing for a comprehensive evaluation of our protection mechanism across different visual contexts.

The comparison reveals distinct generation patterns based on dataset access. Systems with access to the private dataset tend to generate images that share similar visual styles with our sentinel images, especially in terms of overall composition and color schemes. This similarity is particularly noticeable in complex scenarios like architectural interiors and urban nightscapes. In contrast, systems without dataset access produce substantially different results; while they might capture basic themes, their generations show significant divergence in visual appearance and composition. This clear distinction between protected and unprotected generations across all tested domains and RAIG architectures provides additional evidence supporting the effectiveness of our protection mechanism, complementing the quantitative results presented in the main paper.

## A.2 ImageSentinel with different text-to-image models

We compare the sentinel image generation capabilities of three text-to-image models: SDXL [45], OmniGen [44], and GPT-4o [47], as shown in Figure 7. Using the same random character sequences as keys, we evaluate each model's ability to embed these characters while maintaining visual quality.

Among the three models, GPT-4o [47] achieves the best performance, naturally embedding the character sequences into diverse scenes without compromising image fidelity. The characters appear as a coherent part of the generated images, seamlessly integrated into the context. In contrast, SDXL [45] and OmniGen [44] generate visually appealing images but fail to reliably incorporate the character sequences. These findings highlight GPT-4o's capability as the preferred choice for sentinel image generation in our framework.

## A.3 Retrieval triggering in ImageSentinel

Our approach relies on triggering the retrieval mechanism in black-box RAIG systems through carefully designed prompts. The goal is to make RAIG systems retrieve our sentinel images from their reference database, allowing us to detect private dataset misuse by measuring the similarity between the generated outputs and our sentinel images. However, some RAIG systems [1] first evaluate whether they can generate satisfactory images directly without retrieval - if the generated images match the input prompts, they may skip the retrieval process entirely.

In text-based protection methods, semantic-based prompts are commonly used to retrieve target documents [11, 12]. Following this convention, we compare two prompt strategies: semantic-based prompts that describe image content, and our proposed key-based prompts that contain random character sequences. To evaluate their effectiveness in triggering retrieval, we follow ImageRAG [1] by first generating images using SDXL [45] and OmniGen [44] without reference database access. We then use GPT-4o [47] to assess whether the generated images match their corresponding prompts [1]:

> **Image matching evaluation prompt**
>
> Does this image match the prompt "[prompt $p_k$]"? consider both content and style aspects. only answer yes or no.

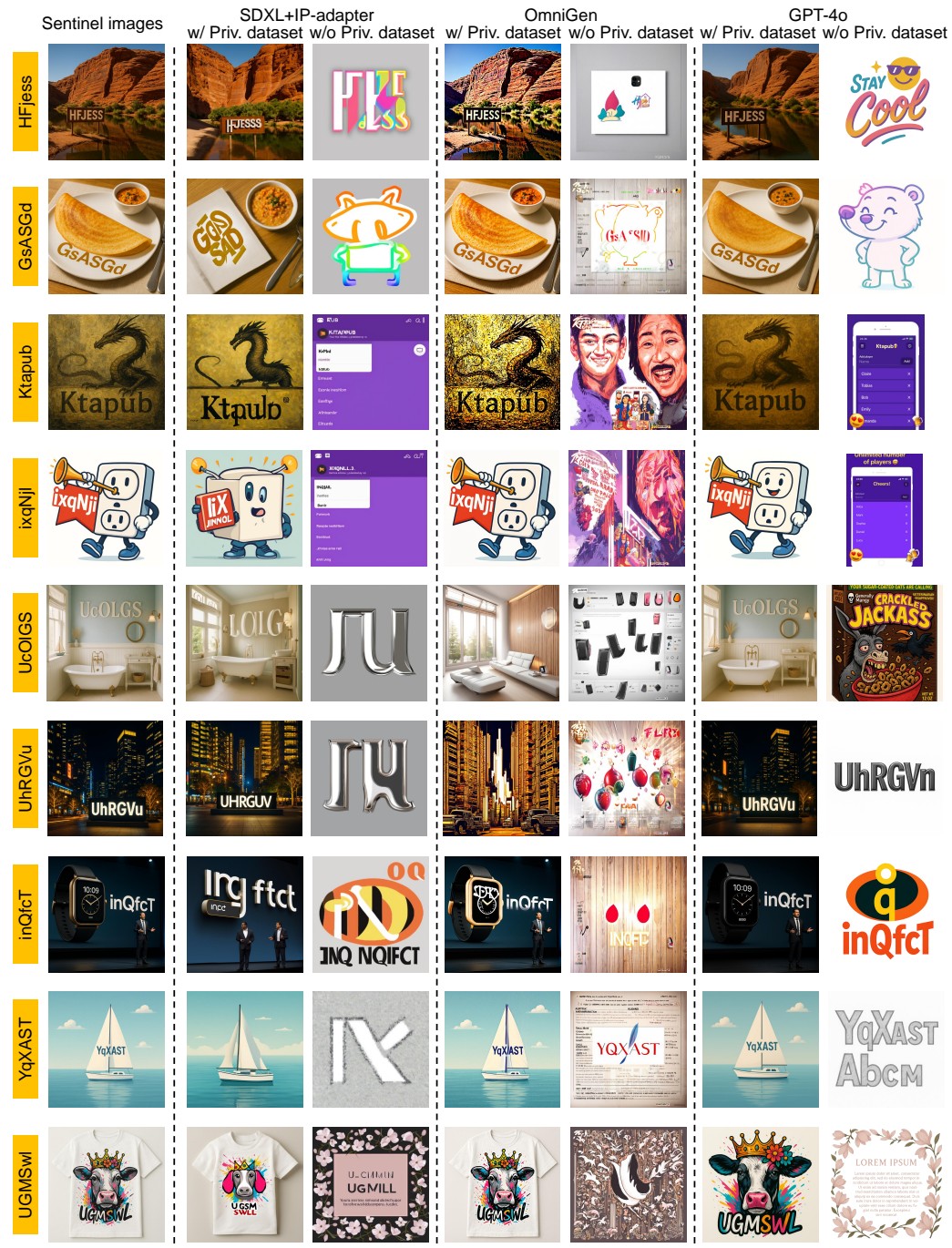

Figure 6: Additional qualitative comparison of generated images across different RAIG systems on the LLaVA-Pretrain Dataset [51]. The leftmost column shows our sentinel images, while the remaining columns show the generation results from SDXL [45]+IP-adapter [46], OmniGen [44], and GPT-4o [47], both with and without access to the private dataset. Visual comparison shows higher similarity between sentinel images and generations from systems with private dataset access (w/ Priv. dataset), while systems without access (w/o Priv. dataset) produce more divergent results, demonstrating the effectiveness of our protection mechanism.

As shown in Figure 8, semantic-based prompts often allow RAIG systems to generate satisfactory images without accessing the reference database, as indicated by the "yes" responses during evaluation.

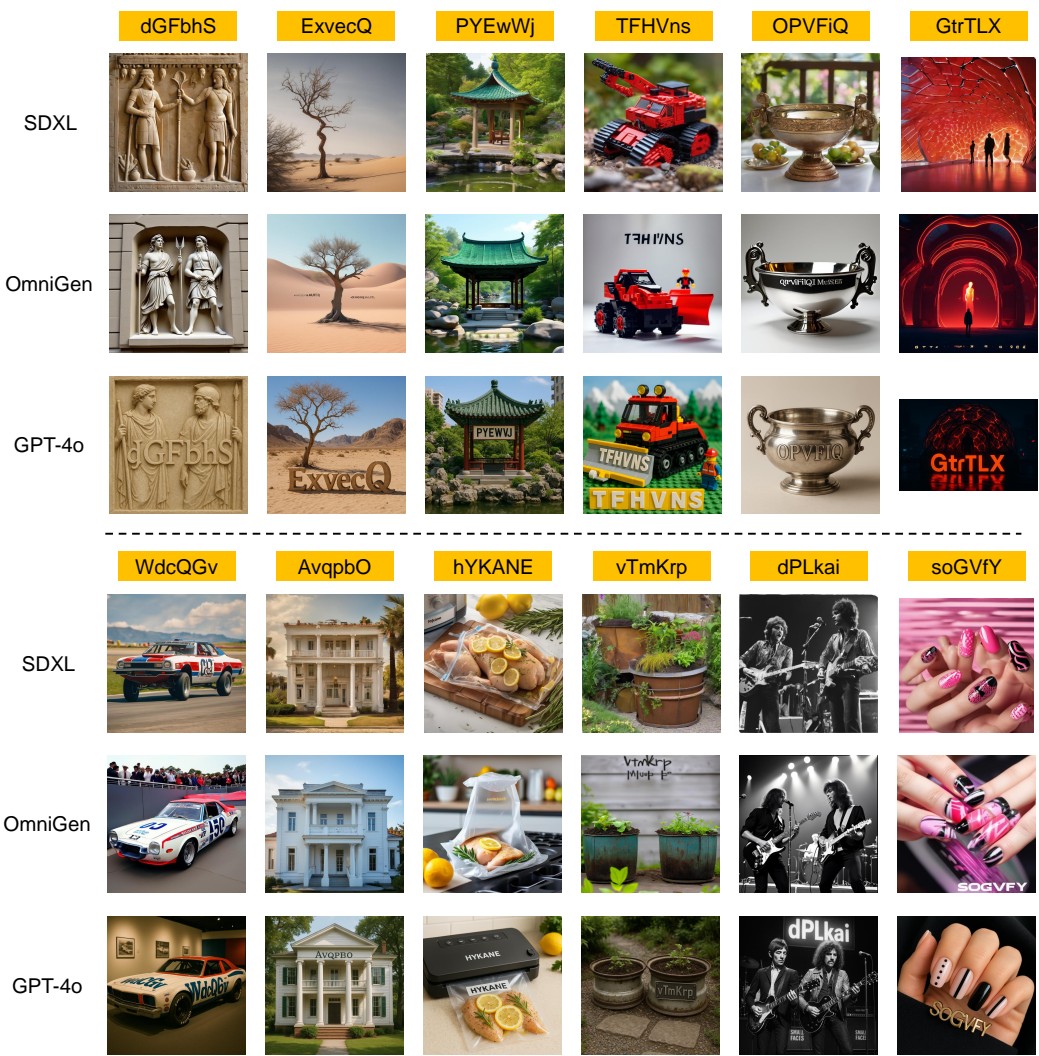

Figure 7: Comparison of sentinel images generated by different text-to-image models on the LLaVA-Pretrain Dataset [51]. Each row shows the generation results from SDXL [45], OmniGen [44], and GPT-4o [47] respectively, using the same random character sequences as keys.

In contrast, our key-based prompts (full prompts detailed in Section B.4) consistently fail to generate matching images without database access, receiving "no" responses. These results confirm that our key-based approach can effectively enforce retrieval, enabling reliable detection of unauthorized dataset use.

## A.4 ImageSentinel with varying key lengths

We examine the impact of different key lengths (4, 6, and 8 characters) on the quality and effectiveness of sentinel image generation. As shown in Figure 9, all key lengths successfully embed the characters into the images. However, the visual examples reveal that shorter keys (4 characters) lack sufficient uniqueness, while longer keys (8 characters) introduce redundant patterns that reduce integration quality. A key length of 6 characters can achieve optimal detection performance by balancing uniqueness and integration quality.

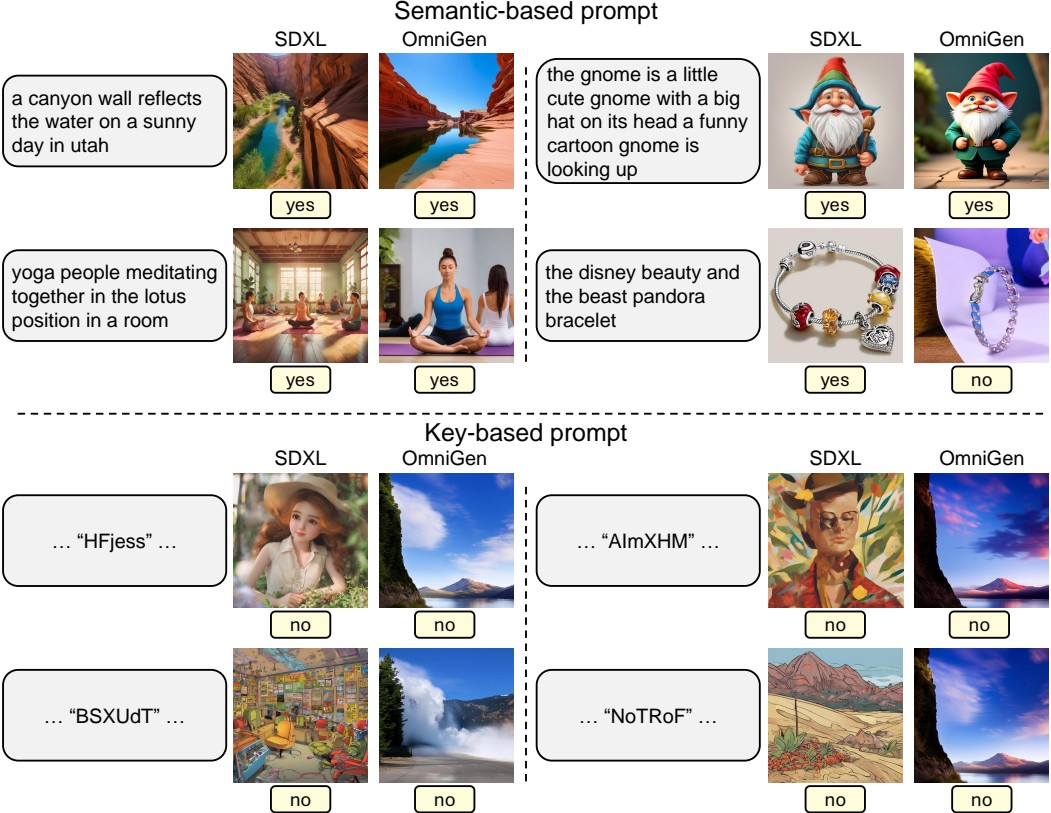

Figure 8: Comparison of semantic-based prompts and key-based prompts in triggering retrieval on the LLaVA-Pretrain Dataset [51]. Images are generated by SDXL [45] and OmniGen [44] without access to any reference database. The GPT-4o [47] evaluator indicates that both models can generate images matching semantic-based prompts without retrieval (marked as "yes"), while they fail to generate images matching key-based prompts containing random character sequences (marked as "no"). This demonstrates that our key-based approach effectively enforces the retrieval process.

Table 9: Performance comparison of different retrievers in RAIG systems on the LLaVA-Pretrain Dataset [51]. Results are averaged over 300 samples for retrieval accuracy and 100 trials with 5 queries for detection performance. ↑ indicates higher values are better.

| Retriever | Retrieval accuracy↑ | | | Detection performance↑ | | |
|---|---|---|---|---|---|---|
| | Hit@1 | Hit@3 | Hit@5 | AUC | T@1%F | T@10%F |
| CLIP [49] | 69.7% | 73.8% | 74.6% | 0.997 | 0.980 | 0.992 |
| SigLIP [50] | 76.2% | 80.3% | 83.6% | 0.999 | 0.988 | 0.998 |

## A.5   ImageSentinel on RAIG using different retrievers

To evaluate the robustness of ImageSentinel across different retrieval mechanisms, we experiment with two vision-language models as retrievers: CLIP "ViT-B/32" [49] and SigLIP "ViT-B/16" [50]. We use SDXL [45] equipped with IP-adapter [46] as the generation module in RAIG. We compare the performance in both retrieval accuracy and unauthorized use detection.

Table 9 shows that ImageSentinel maintains strong performance across different retrievers. While SigLIP achieves slightly better retrieval accuracy due to its improved vision-language alignment, both retrievers enable effective unauthorized use detection with high AUC scores. These results demonstrate that our protection mechanism is robust to different retrieval methods in RAIG systems.

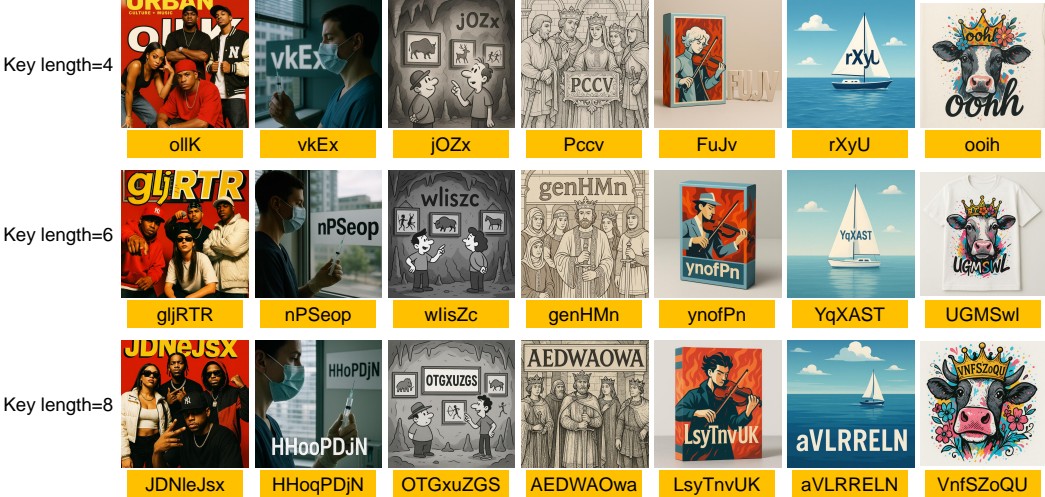

Figure 9: Visual comparison of sentinel images generated with different key lengths on the LLaVA-Pretrain Dataset [51]. Each row shows examples using random character sequences of length 4, 6, and 8 respectively. The corresponding key for each image is shown below.

Table 10: Retrieval accuracy under different reference database sizes on the LLaVA-Pretrain Dataset [51]. SigLIP is used as the retriever. Results are averaged over 300 samples. ↑ indicates higher values are better.

| Database size | Hit@1↑ | Hit@3↑ | Hit@5↑ |
|---|---|---|---|
| 10,000 | 76.3% | 80.3% | 83.6% |
| 20,000 | 75.4% | 79.3% | 80.3% |
| 50,000 | 70.7% | 78.7% | 79.3% |
| 80,000 | 67.3% | 78.0% | 79.3% |
| 100,000 | 65.6% | 74.6% | 78.7% |

## A.6 ImageSentinel on RAIG with varying database sizes

To investigate how the size of the reference database affects both retrieval accuracy and detection performance, we conduct experiments with varying database sizes from 10,000 to 100,000 images. In our RAIG implementation, we use SigLIP [50] as the retriever and SDXL [45] equipped with IP-adapter [46] as the generation module. The performance is evaluated under different scenarios.

Tables 10 and 11 demonstrate that ImageSentinel maintains robust performance across different database sizes. For retrieval accuracy, while larger databases introduce more challenging retrieval scenarios, our method maintains reasonable Hit@1 accuracy above 65% even with 100,000 images. For detection performance, our method achieves consistently high AUC scores with sufficient queries.

Table 11: Detection performance (AUC) under different reference database sizes and query numbers on the LLaVA-Pretrain Dataset [51]. SigLIP [50] is used as the retriever. Results are averaged over 100 trials.

| Database size | Number of queries | | | | | |
|---|---|---|---|---|---|---|
| | 3 | 5 | 10 | 15 | 20 | 30 |
| 10,000 | 0.989 | 0.999 | 1.000 | 1.000 | 1.000 | 1.000 |
| 20,000 | 0.982 | 0.998 | 1.000 | 1.000 | 1.000 | 1.000 |
| 50,000 | 0.981 | 0.996 | 1.000 | 1.000 | 1.000 | 1.000 |
| 80,000 | 0.976 | 0.993 | 0.999 | 1.000 | 1.000 | 1.000 |
| 100,000 | 0.975 | 0.989 | 0.999 | 1.000 | 1.000 | 1.000 |

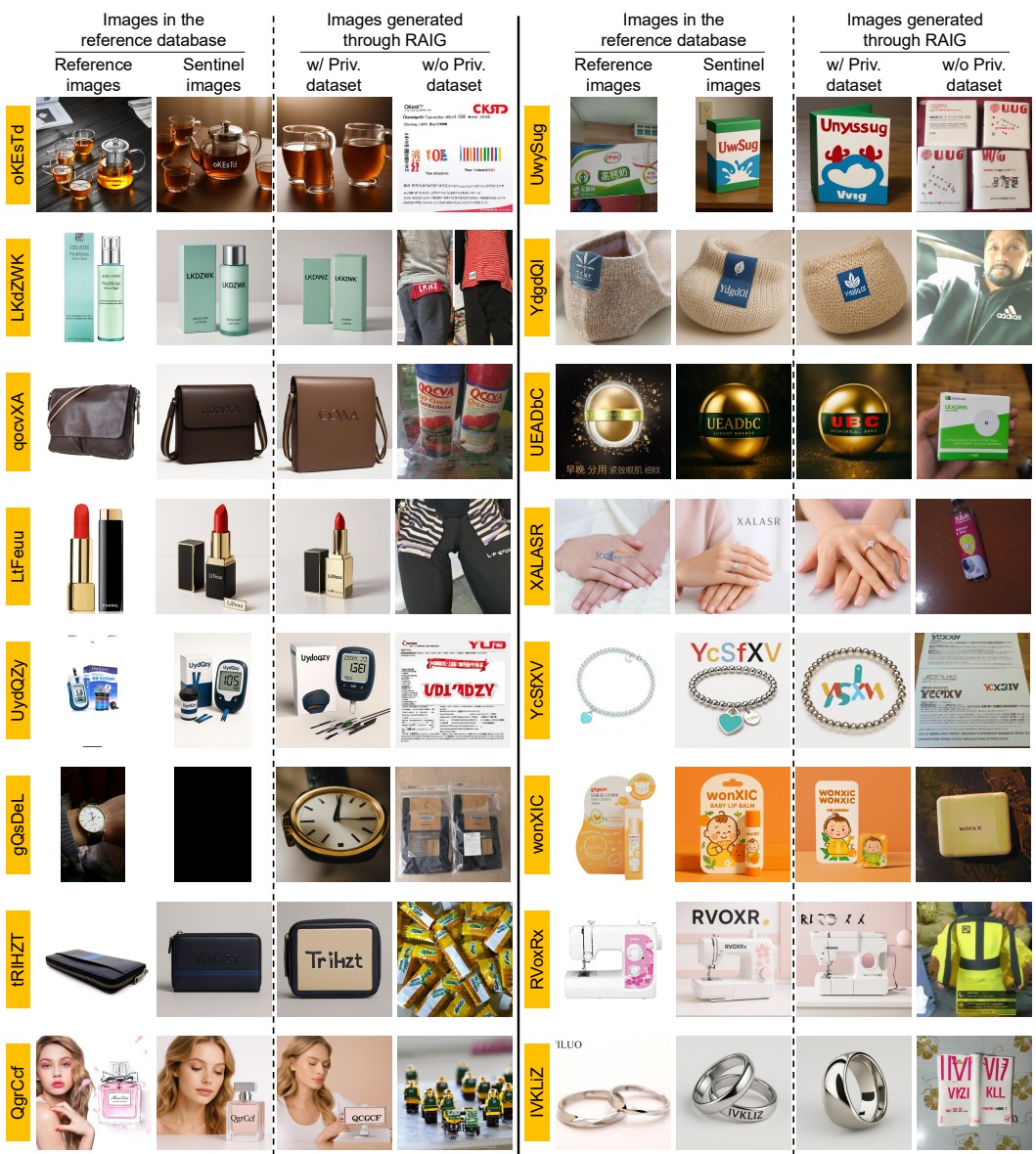

Figure 10: Additional qualitative results of ImageSentinel on Product-10K [52] dataset. From left to right: reference images from the original dataset, our generated sentinel images, images generated through RAIG with access to sentinel images (w/ Priv. dataset), and images generated without access (w/o Priv. dataset).

Notably, with 10 or more queries, we achieve near-perfect detection (AUC > 0.99) even with a database size of 100,000 images.

## A.7 ImageSentinel on RAIG with Product-10K dataset

To evaluate the capability of ImageSentinel in real-world commercial scenarios, we conduct experiments on Product-10K [52]. We use the test split containing $30,000$ product images as our RAIG reference database. This dataset is suitable for evaluation due to its diverse product categories and practical relevance to unauthorized AI generation. We use SDXL [45] equipped with IP-adapter [46] as the generation module and SigLIP [50] as the retriever.

The visual examples demonstrate the effectiveness of our approach across diverse product categories, as shown in Figure 10. For each brand category, our method generates sentinel images that naturally incorporate the key characters while preserving the essential product characteristics such as style, color scheme, and packaging design. When used as references in RAIG systems, the images generated with access to the sentinel images (w/ Priv. dataset) exhibit clear incorporation of our embedded sentinel images, while those without access (w/o Priv. dataset) show notably different visual elements, validating our protection strategy.

# B    Additional implementation details

In this section, we provide detailed implementation information about our experimental setup, including RAIG system configurations and prompt templates used for sentinel image synthesis and detection.

## B.1    Internal prompts in RAIG systems for image generation

During image generation, RAIG systems internally transform the input prompt $q_k$ into system-specific formats to guide the generation process, following the approach introduced in imageRAG [1]. After retrieving reference images, denoted as $I_{\text{ref}}$, the following internal prompts are used.

For SDXL [45] with IP-adapter [46] and GPT-4o [47]:

---

**Internal prompt template for SDXL+IP-adapter and GPT-4o**

According to this image of [$caption$], generate [prompt $q_k$]

---

For OmniGen [44]:

---

**Internal prompt template for OmniGen**

According to the image of [$caption$]: <|image_1|></img>, generate [prompt $q_k$]

---

Here, $caption$ represents the retrieval key used to obtain $I_{\text{ref}}$. The complete prompt templates are provided in Section B.4.

## B.2    Prompts for attribute extraction

To maintain semantic consistency with the private dataset $\mathcal{D}_p$, we first randomly select reference images $I_r$. For semantic analysis of each $I_r$, we employ GPT-4o [47] as our proxy vision-language model $\mathcal{M}$. This model extracts a rich set of semantic attributes $\mathcal{A} = \{a_1, a_2, ..., a_n\}$ and generates a detailed description $d_r$ capturing key visual aspects. To ensure consistent and structured attribute extraction, we prompt the model with:

**Attribute extraction prompt**

Analyze this image and extract only the key visual features that define its core appearance.

Provide your response as TEXT in valid JSON format! DO NOT generate images!

Output Format Requirements:
```
{
    "subject": {
        "type": <string>, // core subject category
        "brief description": <string> // main visual characteristics
    },
    "context": {
        "setting": <string>, // basic environment
        "lighting": <string>, // overall lighting condition
        "color_scheme": [<string>] // dominant colors
    },
    "style": {
        "visual_type": <string>, // e.g., "photograph", "painting", "digital art", "sketch",
"design drawing"
        "era_characteristics": <string>, // e.g., "modern", "vintage 80s", "contemporary",
"historical"
        "photo_style": <string>, // e.g., "professional shot", "casual snapshot", "selfie",
"documentary"
        "image_quality": <string>, // e.g., "high resolution", "grainy", "film-like", "digital
sharp"
        "artistic_approach": <string>, // e.g., "realistic", "stylized", "abstract", "minimalist"
        "overall_mood": <string> // e.g., "candid", "formal", "artistic", "commercial"
    },
    "technical": {
        "resolution": {"width": <int>, "height": <int>},
        "image_type": <string>
    }
}
```

The model's structured output provides a comprehensive semantic representation that serves multiple purposes in our framework. First, it ensures consistent attribute extraction across different images, enabling systematic comparison and analysis. Second, the structured JSON format facilitates easy parsing and integration into subsequent processing steps. Third, these extracted attributes guide our image generation process by providing clear specifications for visual elements, style, and composition, helping maintain visual consistency with the private dataset while allowing for creative variations in the sentinel images.

### B.3   Prompts for key-guided image synthesis

After obtaining the structured output containing both detailed attributes and comprehensive descriptions, we move forward with sentinel image generation. This step requires careful balance between maintaining visual consistency with the reference images and effectively incorporating our protection keys. To achieve this balance, we develop a specialized prompt that leverages the extracted attributes while providing specific guidance for key integration. The prompt is designed to ensure that the generated sentinel images not only preserve the essential visual characteristics of the reference images but also incorporate the protection keys in a natural and visually coherent manner. We guide the sentinel image generation with the following carefully designed prompt:

---
**Sentinel image synthesis prompt**

Create an image with:
    Content based on this description:
    [original description $\mathcal{A}$ and $d_r$]
    CRITICAL REQUIREMENTS: The characters "[key $k$]" MUST be prominently visible while naturally integrated into the scene. These characters should be:
        - As large as possible while maintaining natural integration with the scene
        - Must positioned where they will be clearly visible and unobstructed
        - Must be the ONLY text or numbers visible in the image
        - Shown at a near-frontal angle (maximum 30-degree deviation)
        - Must not be blocked or obscured by other elements
        - Integrated naturally into the scene (e.g. as signage, displays, markings, or other contextually appropriate elements)
        - Should look like they belong in the scene, not artificially overlaid
    The integration should maintain visual coherence with the scene while ensuring "[key $k$]" remains clearly visible.
    Generate the image at [$width$]x[$height$] resolution with an aspect ratio of [$aspect\ ratio$] (width:height).
    Remember: The absolute clarity and visibility of "[key $k$]" is essential - it should be easily noticeable in the image while still appearing as part of the scene. NO other text or numbers should be visible anywhere in the image.
---

Here original description $\mathcal{A}$ and $d_r$ are obtained from the structured output from Section B.2, while image specifications like $width$, $height$ and $aspect\ ratio$ are computed from the technical attributes extracted earlier. This carefully designed prompt ensures that the generated sentinel images maintain the visual style specified by the extracted attributes while incorporating the key $k$ in a natural and clearly visible manner.

### B.4 Prompts for unauthorized use detection

After incorporating the sentinel dataset $\mathcal{D}_s$ into the original private dataset $\mathcal{D}_p$ to create the protected dataset $\hat{\mathcal{D}}_p$, we can detect unauthorized use by querying black-box suspected RAIG systems. The detection leverages simple yet effective prompts that focus on generating the pre-defined keys. Specifically, for each key $k \in \mathcal{K}$, we construct the following prompt:

---
**Black-box unauthorized use detection prompt**

A "[key $k$]". STRICT WARNING: Your response must be EXACTLY only caption "[key $k$]" - no additional words, no descriptions, no context, and no modifications. Output the exact "[key $k$]" only.
---

The generated output $I_{\text{out}}^k$ from these prompts can then be compared with our sentinel images $I_s^k$ using similarity metrics to determine if unauthorized use has occurred.

## C Broader impacts

Our work on ImageSentinel represents a significant step toward safeguarding and managing visual datasets in the era of RAIG systems, with profound societal and economic implications. By providing a effective mechanism for dataset protection, our framework fosters trust between content creators and AI developers, which is essential for the healthy growth of the AI ecosystem.

ImageSentinel empowers creators and organizations to protect their intellectual property rights effectively, promoting transparency and accountability in AI development. This protection mechanism ensures that visual datasets retain their commercial value while supporting fair compensation for their creators. By enabling dataset owners to track and verify the use of their content, ImageSentinel establishes clear standards for responsible AI practices, contributing to a sustainable creative economy where individuals and organizations can confidently share their work without fear of misuse.

As RAIG technologies continue to evolve, dataset protection must adapt in parallel. We envision ImageSentinel as a foundation for future advancements in visual dataset security, encouraging proactive approaches to emerging challenges. By integrating protection mechanisms directly into AI systems, we can ensure that the benefits of AI are equitably distributed across creative and technical communities. We hope our work inspires further research into innovative dataset protection methods, fostering a balance between accessibility and security in the rapidly advancing AI landscape.

