# OpenReview forum: "ImageSentinel: Protecting Visual Datasets from Unauthorized Retrieval-Augmented Image Generation"
_NeurIPS.cc/2025/Conference — NeurIPS 2025 poster_

### Official Review · Reviewer_yaqi · 2025-06-16

**Clarity:** 3
**Significance:** 2
**Originality:** 2
**Rating:** 4
**Confidence:** 4

**Summary:**

The paper aims to detect unauthorized usage of private image datasets in a Retrieval-Augmented Image Generation (RAIG) system. Authors propose to include some digitally crafted images referred as “image sentinels” into the dataset to enable the detection of unauthorized usage. Authors synthesize these sentinels using reference images from the dataset and explicitly condition the image synthesizer to embed a random character sequence (key) into the generated images. These generated images are added to the private dataset as image sentinels. When queried the black-box RAIG system with the keys i.e., specific random character sequences, the detection system compares the RAIG generated images with the image sentinels to find the similarity and raise the detection flag if similarity score exceeds a threshold. Experiments show that the proposed approach reaches 100% detection rate with 10-20 queries to the RAIG system and outperforms watermarking methods Ward-HiDDeN and Ward-FIN.

**Questions:**

I suggest authors to consider the points mentioned in the weakness section to thoroughly evaluate the robustness of the proposed detection mechanism. Currently my rating stands at borderline reject. However if my concerns are adequately addressed during rebuttal, I am willing to reconsider my rating.

**Ethical Concerns:**

["NO or VERY MINOR ethics concerns only"]

**Final Justification:**

This paper addresses an important topic: detecting unauthorized usage of private image datasets in the GenAI era. The idea of enabling detection to be triggered on demand is particularly compelling. During the discussion, the authors provided a comprehensive analysis of critical adaptive attacks, which is appreciated and offers a thorough understanding of the proposed protection. After the rebuttal, my concerns regarding adaptive attacks were resolved, and I find the work convincing and valuable for the community to learn about.

**Limitations:**

Authors have not explored possible detection evading mechanism to counter their proposed method.

**Quality:**

2

**Strengths And Weaknesses:**

Strengths:

-	A simple but shown to be an effective approach to detect unauthorized image of private image datasets.

-	No modifications are made to the original images in the dataset, hence authorized usage still provides relevant results.

-	These sentinels make the detection to be triggered on demand i.e., reveal only when explicitly prompted with the specific keys (random character sequence).

-	It is a simple and straightforward pipeline to generate the image sentinels leveraging existing vision-language foundation models, and find image similarity using vision foundation models at the time of detection.

-	Results show that the detection rate can be reached to 100% with 10-20 queries, outperforming watermarking based methods.

-	Paper is clearly written, and it is easy to read and understand.

Weakness:
Despite the strengths mentioned above, authors have not explored any possible detection evading techniques. I find that is the major weakness of the paper. A RAIG provider could easily incorporate some simple techniques to remove the image sentinels from the dataset:

-	Identifying duplicate images in the dataset, and finding representative images that are unique based on the type, description, characteristics or even with perceptual similarity using DINO and filtering the duplicates or over-represented content.

-	Another approach could be leveraging OCR models to detect the text from images and filtering out images with random character sequences or masking such image regions.

---

> ### Author Rebuttal · Authors · 2025-07-29
>
> We appreciate Reviewer yaqi's insightful feedback and specific suggestions.
>
> > ### **[W1] Identifying duplicate images in the dataset, and finding representative images that are unique based on the type, description, characteristics or even with perceptual similarity using DINO and filtering the duplicates or over-represented content.**
>
> Thank you for raising this important concern about duplicate content filtering. We conducted additional experiments to evaluate this potential attack strategy.
>
> **1. Attack Implementation**
>
> We implemented a duplicate filtering attack using DINO similarity scores. For each image pair, we compute their DINO similarity and remove images exceeding a threshold $\delta$, prioritizing the removal of images containing random character-like patterns.
>
>
> **2. Performance**
>
> We evaluated various filtering thresholds $\delta$ (0.5-0.9) across two datasets with 10 queries for detection:
>
> LLaVA Visual Instruct Dataset:
> | $\delta$ | AUC | TPR@1%FPR | TPR@10%FPR | Clean Images Filtered Rate |
> |---|-----|-----------|------------|------------|
> |0.5|0.88 |0.58      |0.77        |87.9%|
> |0.6|0.98 |0.92      |0.97        |62.8%|
> |0.7|1.00 |0.98      |1.00        |34.2%|
> |0.8|1.00 |1.00      |1.00        |13.1%|
> |0.9|1.00 |1.00      |1.00        |3.3%|
>
> Product-10K Dataset:
> | $\delta$ | AUC | TPR@1%FPR | TPR@10%FPR |Clean Images Filtered Rate|
> |---|-----|-----------|------------|------------|
> |0.5|0.90 |0.60      |0.78        |96.2%|
> |0.6|0.98 |0.87      |0.95        |83.1%|
> |0.7|1.00 |0.98      |0.99        |60.2%|
> |0.8|1.00 |1.00      |1.00        |37.3%|
> |0.9|1.00 |1.00      |1.00        |19.0%|
>
> **3. Analysis:**
>
> - Lower $\delta$ thresholds (0.5-0.6) can filter out sentinel images and cause low detection accuracy.
> - However, they lower $\delta$ thresholds also remove an unacceptable proportion of legitimate images (62.8-96.2%). This severely compromises the dataset's utility. Especially for Product-10K dataset, it suffers more severe filtering due to its inherent nature of containing visually similar product images.
> - Setting a high $\delta$ (0.8-0.9) preserves more legitimate content but fails to effectively filter sentinel images, as evidenced by the high detection scores.
> - Our method's resilience against similarity-based filtering is largely attributed to our pipeline shown in Figure 3. Instead of directly feeding reference images into GPT-4o for sentinel image generation, we first perform attribute extraction followed by language-guided generation. Direct reference-guided generation would result in substantial content overlap, while our language-guided approach maintains stylistic consistency while avoiding identical content duplication. This design ensures that sentinel images share semantic and stylistic elements with the reference images while maintaining sufficient visual diversity to survive similarity-based filtering.
>
> We will include these quantitative results and visual examples in our revised paper to better illustrate our findings.
>
> > ### **[W2] Another approach could be leveraging OCR models to detect the text from images and filtering out images with random character sequences or masking such image regions.**
>
> We appreciate this insightful comment about detection evading techniques. We have conducted experiments to evaluate potential attacks that attempt to detect and neutralize sentinel images.
>
> **1. Attack Implementation**
>
> We implemented two attack strategies using OCR-based random character detection:
> - Filtering: removing detected sentinel images
> - Masking: obscuring detected text regions
>
> **2. OCR Detection Performance**
>
> We use three widely-used OCR systems (EasyOCR, Tesseract OCR, and PaddleOCR) on two datasets, and apply the PyEnchant library [1] to determine whether the detected text contains random character sequences. The results are shown below.
>
> | OCR System | Dataset | Sentinel Images Detection Rate | Clean Images Filtered Rate |
> |------------|---------|----------------|----------------------|
> | EasyOCR | Product-10K | 92.7% | 41.7% |
> | EasyOCR | LLaVA Visual Instruct | 90.2% | 35.3% |
> | Tesseract OCR | Product-10K | 63.6% | 39.7% |
> | Tesseract OCR | LLaVA Visual Instruct | 61.5% | 23.7% |
> | PaddleOCR | Product-10K | 98.3% | 51.1% |
> | PaddleOCR | LLaVA Visual Instruct | 97.5% | 38.0% |
>
> While OCR-based attacks are possible to filter most sentinel images, the **high false positive rates** on clean images indicate that such attacks would significantly impair RAIG's normal functionality. This is particularly problematic for datasets like Product-10K, where a substantial portion of legitimate images containing product names, URLs, or watermarks would be incorrectly filtered out.
>
> [1] Ryan, K. PyEnchant: a spellchecking library for Python.
>
> **3. Impact Analysis**
>
> Detection performance under EasyOCR-based attacks (LLaVA Visual Instruct dataset):
>
> | Queries | Attack Method | AUC | TPR@1%FPR | TPR@10%FPR |
> |---------|--------------|------|------------|------------|
> | 5       | No Attack    | 0.99 | 0.98       | 0.99       |
> |         | Filtering    | 0.63 | 0.16       | 0.34       |
> |         | Masking      | 0.62 | 0.15       | 0.34       |
> | 10      | No Attack    | 1.0  | 1.0        | 1.0        |
> |         | Filtering    | 0.71 | 0.19       | 0.40       |
> |         | Masking      | 0.69 | 0.20       | 0.42       |
> | 50      | No Attack    | 1.0  | 1.0        | 1.0        |
> |         | Filtering    | 0.90 | 0.67       | 0.80       |
> |         | Masking      | 0.91 | 0.65       | 0.82       |
> | 100     | No Attack    | 1.0  | 1.0        | 1.0        |
> |         | Filtering    | 0.97 | 0.93       | 0.96       |
> |         | Masking      | 0.98 | 0.94       | 0.96       |
>
> Generation quality impact (300 samples):
>
> | Attack Method | CLIP | SigLIP | DINO |
> |--------------|-------|----------|--------|
> | No Attack    | 0.772 | 0.743    | 0.605  |
> | Filtering    | 0.531 | 0.527    | 0.503  |
> | Masking      | 0.559 | 0.546    | 0.511  |
>
> - These results demonstrate that while attacks can initially degrade detection effectiveness, this can be mitigated by increasing query numbers. For example, under filtering attacks, the AUC drops to 0.63 with 5 queries, but recovers to 0.97 with 100 queries.
> - Such attacks significantly compromise RAIG's generation quality, making them impractical for real-world deployment. Under attacks, the CLIP scores drop from 0.772 to around 0.531 (Filtering attack) and 0.559 (Masking attack), while SigLIP scores decrease from 0.743 to around 0.53 (Filtering attack) and 0.55 (Masking attack), representing a degradation in generation quality.
>
> We will discuss these potential detection evading techniques in an expanded limitations section, along with detailed experimental results. Future work could explore more sophisticated attack strategies and corresponding defense mechanisms to further validate the robustness of our protection framework.

---

> > ### Author Response · Authors · 2025-08-06
> >
> > Dear Reviewer yaqi,
> >
> > Thank you for dedicating your time and effort to reviewing our paper. We have conducted additional experiments addressing your concerns about duplicate filtering attacks and OCR-based detection attacks. These analyses have helped us better understand the robustness of our approach.
> >
> > If you have any follow-up questions about our work, we welcome further discussion.
> >
> > We sincerely appreciate your efforts, which have significantly contributed to improving our work.
> >
> > Best regards,
> >
> > The Authors

---

> ### Comment · Reviewer_yaqi · 2025-08-06
>
> Thank you for running the adaptive attacks. I have a few clarifying questions and suggestions:
>
> 1. **Clean-image filtered:** When authors report that X % of “clean” images are filtered out, are those images share the same attribute values as the retained images? It would be helpful to measure the attribute-space coverage of the surviving dataset—ideally using the same attribute taxonomy you employed when synthesizing sentinels. This would show whether the remaining images still form a representative sample of the original distribution. Reporting only the raw percentage removed makes it hard to judge the real utility loss. This coverage analysis applies to both the DINOv2-similarity filter and the OCR-based detection.
>
> 2. **Text masking / rewriting:** In practice, a RAIG provider could
>
> - **Offline phase:** detect and in-paint text regions, or regenerate text-free variants with an image-to-image diffusion model before building the retrieval index.
>
> - **Online phase:** If the retrieved image still contains text, perform a lightweight second-pass in-painting (or apply a targeted “rewrite-the-text” diffusion prompt), then repeat retrieval to ensure the result matches the query description before returning the final output.
>
> These steps allow the RAIG system to preserve retrieval quality while neutralizing sentinel keys.
>
> 3. **Masking evaluation:** When evaluating the performance of masking, different queries were used. However, such masking could be performed by the RAIG provider to refine the dataset before it is indexed for retrieval.
>
> Overall, I appreciate the thorough evaluation, but I believe an explicit attribute-coverage metric would clarify the true loss of dataset utility, and RAIG providers could automate text removal or rewriting both offline and online.

---

> > ### Author Response · Authors · 2025-08-07
> > **Our Response (1/2)**
> >
> > Thank you for your insightful questions. We have conducted additional experiments and analyses to address your concerns.
> >
> > **1.1. Clean-image false positives: When you report that X % of “clean” images are filtered out, are those images near-duplicates or instances that share the same attribute values as the retained images?**
> >
> > We evaluated the filtered images from Products-10K dataset both quantitatively and qualitatively. For quantitative analysis, we used GPT-4.1 mini to determine whether the filtered images are near-duplicates or share similar attributes with the retained images. The prompt used was:
> >
> > ```
> > Please analyze these two images and answer with ONLY "Yes" or "No":
> > Are these two images either near-duplicates (very similar visual content) OR do they share highly similar attributes (same subject type, style, and composition)?
> >
> > Rules:
> > 1. Near-duplicates means: same scene/object from similar angle with minimal variations
> > 2. Similar attributes means: same subject type + same visual style + similar composition
> > 3. Only answer "Yes" if either condition is met
> > 4. Only answer "No" if neither condition is met
> > 5. Do not explain your reasoning, only answer Yes/No
> > ```
> >
> > We examined 500 filtered images (threshold=0.6) and found that only 34.0% were identified as similar (answered "Yes"), while 66.0% were deemed dissimilar. This indicates that DINO similarity-based filtering removes many images beyond just near-duplicates or instances sharing similar attributes.
> >
> > Qualitatively, we examined the filtered images and found that DINO similarity-based filtering leads to many false positives. For example:
> > - 1001147.jpg (white bag) vs 4312032.jpg (white chef's hat)
> > - 9839346.jpg (pink baby clothes) vs 7828668.jpg (black sports long-sleeve)
> > - 8006534.jpg (bottled shampoo) vs 9717137.jpg (canned plant powder)
> >
> > We also discovered that GPT-4.1 mini sometimes struggles with accurate similarity judgments. For instance, with 1000472.jpg (white pillow) and 1065085.jpg (similar-shaped blue pillow with cartoon patterns). We believe each of these examples has unique properties valuable for Retrieval-Augmented Image Generation. We will include representative visual examples in the revised paper.
> >
> >
> > **1.2. It would be helpful to measure the attribute-space coverage of the surviving dataset—ideally using the same attribute taxonomy you employed when synthesizing sentinels.**
> >
> > Thank you for the suggestion about measuring attribute-space coverage. We want to clarify that in our sentinel synthesis pipeline, we use GPT-4o for open-ended attribute extraction rather than classification into predefined categories. The extracted attributes are diverse and dataset-specific, making it challenging to directly measure coverage using the same taxonomy.
> >
> > However, we leveraged the existing classification system in Products-10K dataset to analyze the impact on class distribution:
> >
> > | Filter Method      | Parameter | Unique Classes | Class Retention (%) |
> > |-------------------|-----------|----------------|-------------------|
> > | Original Dataset  | -         | 6,574          | 100%             |
> > | DINO-similarity Attack| $\delta$=0.5     | 99             | 1.51%            |
> > |                   | $\delta$=0.6     | 883            | 13.43%           |
> > |                   | $\delta$=0.7     | 2,479          | 37.71%           |
> > |                   | $\delta$=0.8     | 4,010          | 61.00%           |
> > |                   | $\delta$=0.9     | 5,163          | 78.54%           |
> > | OCR-based Attack  | -         | 4,350          | 66.17%           |
> >
> > The results reveal severe disruption to the class distribution. This significantly impacts retrieval-augmented image generation, as different instances may serve unique roles as reference images, each potentially representing unique brands, styles, and colors.
> >
> > We agree that raw percentage removal alone is insufficient to judge real utility loss. In our previous response, we also evaluated generation quality [1] using the dataset after OCR-based attacks as a metric for real utility loss. For DINO-similarity attacks, we did not test this metric because at effective attack thresholds ($\delta$=0.5-0.6), the FPR was already very high (83.1-96.2%), making the reference dataset unusable.
> >
> > [1] Shalev-Arkushin, Rotem, et al. "ImageRAG: Retrieval-Augmented Image Generation with Semantic Guidance." arXiv preprint 2025.

---

> ### Author Response · Authors · 2025-08-07
> **Our Response (2/2)**
>
> **2.1. Offline phase: detect and in-paint text regions, or regenerate text-free variants with an image-to-image diffusion model before building the retrieval index.**
>
> Following your suggestion, we implemented an attack using EasyOCR+Stable Diffusion 2.0 inpainting (**Inpainting Attack**). We evaluated the generation quality after applying this attack on a subset of the LLaVA dataset (1000 images). The results are as follows:
>
> | Attack Method | CLIP | SigLIP | DINO |
> |--------------|-------|----------|--------|
> | No Attack    | 0.788 | 0.762    | 0.653  |
> | Inpainting Attack | 0.785 | 0.752    | 0.645  |
>
> The results show **minimal impact** on generation quality, indicating that this attack can maintain generation performance while potentially compromising protection. Results from larger-scale dataset testing will be included in the next version of the paper.
>
> We greatly appreciate this suggestion, as it highlights what we consider to be a more effective attack method. Yet it's worth noting that this attack approach carries a limitation: important text elements such as brand names might be mistakenly identified and removed, which would impair legitimate brand-based retrieval and generation. While simply increasing the number of sentinel images could help, as shown in our last response, more sophisticated protection mechanisms would be needed to defend against such attacks effectively.
>
> **2.2. Online phase: if a retrieved or generated image still contains text, run a lightweight second-pass in-painting step (or a targeted "rewrite-the-text" diffusion prompt) before returning the final output.**
>
> This proposed attack strategy would be ineffective against our protection mechanism. Our detection mechanism does not rely on rendering text in the generated images. Instead, we detect unauthorized use by measuring **visual similarity** between RAIG-generated images and our sentinel images. While text plays a role during the retrieval phase, post-retrieval text manipulation cannot evade detection because our generation phase only requires the semantic content of retrieved images to generate outputs containing sentinel visual elements, which we then use for detection.
>
> **3. Masking evaluation: When evaluating the performance of masking, different queries were used. However, such masking could be performed by the RAIG provider to refine the dataset before it is indexed for retrieval.**
>
> We would like to ensure we have correctly understood your concern. In our experiments, we did process the dataset before generation, including both the original images and embedded sentinel images. The different queries mentioned in our evaluation refer to the detection process, where each query represents one instance of inputting our designed key to the black-box RAIG system to obtain output images.
>
> We deeply appreciate your suggestions and acknowledge the critical importance of designing effective attacks. These potential risks should be communicated to readers and users to ensure transparency. We also believe understanding vulnerabilities is essential for developing more robust protection mechanisms. We greatly value this discussion and welcome any further questions or concerns during this discussion phase.

---

> ### Comment · Reviewer_yaqi · 2025-08-07
>
> Thank you for the prompt response. Based on the additional experiments and clarifications, I have few comments and suggestions:
>
> **Attribute-space coverage:** I will accept the results with existing classification system in Products-10K dataset to analyze the impact on class distribution for the attribute coverage aspect. This result make it clear that the similarity based approaches fail to retain useful and unique characteristics images in the dataset, and this is not an effective attack strategy for this protection.
>
> **Detect-and-in-paint strategy:** I appreciate the new experiment demonstrating that automatically detecting and in-painting text regions can neutralize sentinel images while preserving generation quality. Expanding this test to a larger scale in the next revision would strengthen the paper.
>
> **Selective text removal:** Authors note that removing brand names can be undesirable. It may be worth acknowledging that an adversary could maintain a vocabulary of critical terms like brand names to avoid erasing them. It is an incremental yet realistic evasion tactic.
>
> **Online phase:** I agree with your statement. I realized this aspect earlier today and edited my comment accordingly. Nevertheless, the same response from offline phase can also apply in this scenario.
>
> **Masked evaluation:** The explanation you provided for the masked evaluation protocol is clear and addresses my question.
>
> Overall, the “detect-and-in-paint” approach appears to be the most credible attack strategy in a grey-box scenario where the protection method is disclosed. My evaluation is not based on whether the protection is easily evaded but to openly discuss such limitations to enhance paper’s transparency and practical value. I believe that including this attack analysis in the revised version would strengthen the contribution.
>
> With these points satisfactorily covered, I will revise my recommendation to borderline accept.
>
> Thank you again for your thorough and thoughtful engagement.

---

> > ### Author Response · Authors · 2025-08-08
> > **Thanks for your constructive feedback**
> >
> > We sincerely appreciate your detailed feedback and suggestions. Thank you for suggesting the effective detect-and-in-paint strategy, and we will expand this analysis in our revision with larger-scale evaluation. We also agree that vocabulary-based text preservation for brand names is a practical solution, and we will incorporate this discussion in our analysis.
> >
> > Thank you again for your guidance in strengthening our paper.
> >
> > Best regards,
> >
> > The Authors

---

### Official Review · Reviewer_qQTS · 2025-07-02

**Clarity:** 3
**Significance:** 4
**Originality:** 3
**Rating:** 5
**Confidence:** 4

**Summary:**

The paper introduces a method to protect image datasets in the context of retrieval augmented generation. It works by generating “sentinel images” and including them in the protected dataset. They are images that incorporate a random-character key in the image, associated with a particular image semantics. To detect the use of a protected dataset, one has to prompt the model with the specific key and analyze the semantic similarity between the sentinel and the generated image (done with DINO). The paper validates the approach on SOTA generative models (GPT-4o, SD-XL, OmniGen) and thorough experiments.

**Questions:**

(1) The paper mentions a total of 10,000 images, with 2,800 images designated as the private dataset. Could the authors clarify the precise number or proportion of these 2,800 private images that are converted into or replaced by sentinel images? If a significant portion, or indeed all, of the 2,800 images serve as sentinels, what are the implications for the probability of inadvertently observing a sentinel image when inspecting the dataset, and what is the potential impact on detection efficiency or overhead?

(2) When the generative model operates on a protected dataset but without the specific retrieval key included in the prompt, has it been observed that the model inadvertently generates the embedded random character sequences within the output images?

**Ethical Concerns:**

["NO or VERY MINOR ethics concerns only"]

**Final Justification:**

The rebuttal has addressed most of my interrogations. It particularly introduced new attacks and additional details. I strongly believe that this paper has its place at the conference and in the field.
Although, the idea of sentinel images in itself leads to strong limitations (no theoretical guarantees on the detection, ease of protection removal), it could serve as a first baseline for protection of image datasets in the context of RAIG and opens up interesting problems to solve. This is why I recommend acceptance.

**Limitations:**

Yes

**Quality:**

3

**Strengths And Weaknesses:**

## Strengths

(1) The paper addresses an important and currently underexplored problem: protecting visual datasets from unauthorized use in Retrieval-Augmented Image Generation (RAIG).

(2) The paper is well-written and easy to understand. It uses good illustrations, like prompts in boxes, which helps in the explanation.

(3) The main idea of using sentinel images to get a clear detection signal is simple and effective. This makes it easy to tell when protected data has been used by a RAIG model.

(4) The experiments are done carefully and thoroughly:
    (a) The code is clean, and seems easy to check and run (although I did not).
    (b) The authors used adapted WARD as a baseline, which, even if different from their method, provides a good comparison point and shows effort to be rigorous.
    (c) The method is tested with several up-to-date generative models (like GPT-4o, SD-XL, and Omnigen), showing its wide applicability.
    (d) Extra results are included in the supplementary materials, adding more proof for their findings.

## Weaknesses

(1) The use of DINO for semantic similarity in detection limits how strong the statistical proof can be. For example, evaluating only TPR at FPR=0.01 might not be enough for legal situations, unlike methods such as the adapted WARD-Hidden, which can offer more statistical tests based on the matching its between binary messages.

(2) Some parts of the experimental setup are not fully clear in the paper:
    (a) The exact number of sentinel images used in the experiments is not explicitly stated, which makes it hard to understand the scale of the protection and how stealthy it is.
    (b) How TPR at 1% FPR is calculated is not completely clear. The paper mentions "100 trials with 5 queries" but doesn't fully explain what a "trial" is or how the "negatives" (without the protected dataset) are created for evaluation.

(3) The paper does not discuss how people might try to get around the protection.
    (a) For example, if a generative model provider sees unusual images with random characters in their database, they might simply remove them.
    (b) They could also prevent users from using prompts with random characters. The paper would benefit from discussing these potential challenges and how the method might handle them.

---
Note for the ratings:
- Quality: the experiments are done with care, but the idea of sentinel images in itself leads to strong limitations (no theoretical guarantees on the detection, ease of protection removal

---

> ### Author Rebuttal · Authors · 2025-07-30
>
> We thank Reviewer qQTS for the thorough review and constructive suggestions.
>
> >### **[W1] Statistical Strength of Detection Evidence**
>
>  We appreciate the suggestion to strengthen our statistical evidence and have conducted additional analyses.
>
> **1. Extended Evaluation Metrics**
>
> We follow practices in detection evaluation [1-4] to use TPR@1%FPR in our experiments. We acknowledge the need for more stringent statistical guarantees for legal applications. We extended our evaluation to include TPR@0.1%FPR, providing stronger statistical confidence aligned with recent works [5,6].
>
> **2. Results**
>
> We keep the same experimental setting as Table 3 in our paper.
>
> | RAIG | Query | AUC | TPR@0.1%FPR | TPR@1%FPR |
> |------|-------|-----|--------|-------|
> | SDXL | 3 | 0.974|0.802|0.934|
> |      | 5 | 0.997|0.952|0.980|
> |      | 10 |1.000|0.998|1.000|
> |      | 20 |1.000|1.000|1.000|
> | OmniGen | 3 | 0.974|0.520|0.584|
> |      | 5 | 0.997|0.632|0.736|
> |      | 10 |1.000|0.880|0.922|
> |      | 20 |1.000|0.992|0.996|
> | GPT-4o | 3 | 0.983|0.863|0.954|
> |      | 5 | 0.997|0.973|0.980|
> |      | 10 |1.000|0.999|1.000|
> |      | 20 |1.000|1.000|1.000|
>
> Our method maintains high detection rates even at FPR=0.001.
>
> We will expand Table 3 in the main paper to include these more stringent metrics.
>
>
> [1] Wen, Y. et al. "Tree-ring watermarks: Fingerprints for diffusion images that are invisible and robust." NeurIPS 2023.
>
> [2] Dathathri, S. et al. "Scalable watermarking for identifying large language model outputs." Nature 2024.
>
> [3] Pan, L. et al. "MarkLLM: An Open-Source Toolkit for LLM Watermarking." EMNLP 2024.
>
> [4] Liu, Y. et al. "Dataset Protection via Watermarked Canaries in Retrieval-Augmented LLMs." ICML 2025 Workshop R2-FM.
>
> [5] Pang, Q. et al. "No free lunch in LLM watermarking: Trade-offs in watermarking design choices." NeurIPS 2024.
>
> [6] Kirchenbauer, J. et al. "On the Reliability of Watermarks for Large Language Models." ICLR 2024.
>
>
> >### **[W2.(a), Q1] Exact number of sentinel images**
>
> In our experiments, as shown in Table 3, S3, and S4, the number of queries is equal to the number of sentinel images used, as each query is designed to trigger a specific sentinel image.
>
> Specifically, our experiments used:
> * Table 3: 3-20 sentinel images in 10,000 references (0.03%-0.2%)
> * Table S3: 3-30 sentinel images in 10,000-100,000 references (0.003%-0.3%)
> * Table S4: 1-20 sentinel images in 30,000 references (0.003%-0.07%)
>
> These sentinel images were randomly selected from a pool of 2,800 private images. The small ratios help ensure the stealthiness of our protection mechanism. We will explicitly clarify these numbers in the revised paper.
>
> >### **[W2.(b)]** Clarity of Performance Metrics Calculation
>
> We explain the process of our trials and TPR/FPR calculation as follows.
>
> **Trial Process**
>
> For each trial with N queries:
>
> 1. Positive Sample Generation:
> - Randomly select N images from our private dataset.
> - Convert them into sentinel images.
> - Embed them into the reference database.
> - Generate N images using their corresponding N key queries.
> - Calculate the mean DINO similarity between generated images and their corresponding sentinel images.
>
> 1. Negative Sample Generation:
> - Use the same N queries but with the original reference database (without sentinel images).
> - Generate N images and calculate mean DINO similarity similarly.
>
> **Evaluation Details**
> - We conduct 100 trials for positive samples.
> - For robust threshold estimation, we conduct 2000 trials for negative samples.
> - TPR@1%FPR is calculated by:
>   1. Setting detection threshold at 1% percentile of negative samples' similarities.
>   2. Computing the fraction of positive samples exceeding this threshold.
>
> We will add these detailed evaluation settings to our revised paper.
>
> >### **[W3.(a)] Vulnerability to Database Filtering**
>
> We have conducted experiments to evaluate potential attacks that attempt to detect and filter sentinel images.
>
> **1. Attack Implementation**
>
> We implemented two attack strategies using OCR-based random character detection:
> - Filtering: removing detected sentinel images.
> - Masking: obscuring detected text regions.
>
> **2. OCR Detection Performance**
>
> We use three widely-used OCR (EasyOCR, Tesseract OCR, and PaddleOCR) on two datasets, and apply the PyEnchant library [1] to determine whether the detected text contains random character sequences. The results are shown below.
>
> |OCR System|Dataset|SentinelImages Detection Rate|Clean Images Filtered Rate|
> |-|-|-|-|
> |EasyOCR|Product-10K|92.7%|41.7%|
> |EasyOCR|LLaVA Visual Instruct|90.2%|35.3%|
> |Tesseract|Product-10K|63.6%|39.7%|
> |Tesseract|LLaVA Visual Instruct|61.5%|23.7%|
> |PaddleOCR|Product-10K|98.3%|51.1%|
> |PaddleOCR|LLaVA Visual Instruct|97.5%|38.0%|
>
> While OCR-based attacks are possible to filter most sentinel images, the **high false positive rates** on clean images indicate that such attacks would significantly impair RAIG's normal functionality. This is particularly problematic for datasets like Product-10K, where a substantial portion of legitimate images containing product names, URLs, or watermarks would be incorrectly filtered out.
>
> [1] Ryan, K. PyEnchant: a spellchecking library for Python.
>
> **3. Impact Analysis**
>
> Detection performance under EasyOCR-based attacks (LLaVA dataset):
>
> |Queries|Attack Method|AUC|TPR@1%FPR|TPR@10%FPR|
> |-|-|-|-|-|
> |5|No Attack|0.99|0.98|0.99|
> ||Filtering|0.63|0.16|0.34|
> ||Masking|0.62|0.15|0.34|
> |10|No Attack|1.0|1.0|1.0|
> ||Filtering|0.71| 0.19|0.40|
> ||Masking|0.69|0.20|0.42|
> |50|No Attack|1.0|1.0|1.0|
> ||Filtering|0.90|0.67|0.80|
> ||Masking|0.91|0.65|0.82|
> |100|No Attack|1.0|1.0|1.0|
> ||Filtering|0.97|0.93|0.96|
> ||Masking|0.98|0.94|0.96|
>
> Generation quality impact (300 samples):
>
> |Attack Method|CLIP|SigLIP|DINO|
> |-|-|-|-|
> |No Attack|0.772|0.743|0.605|
> |Filtering|0.531|0.527|0.503|
> |Masking|0.559|0.546|0.511|
>
> These results demonstrate that while attacks can initially degrade detection effectiveness, this can be mitigated by increasing query numbers. However, such attacks significantly compromise RAIG's generation quality, making them impractical for real-world deployment.
>
> We will discuss these potential adaptive adversaries in an expanded limitations section, along with detailed experimental results. Future work could explore more sophisticated attack strategies and corresponding defense mechanisms to further validate the robustness of our protection framework.
>
> >### **[W3.(b)] Vulnerability to Prompt Filtering**
>
> We have conducted additional experiments to evaluate our approach's robustness against prompt preprocessing:
>
> **1. Prompt Stripping Attack**
>
> We implemented a **prompt stripping** attack that removes random character sequences in the queries.
>
> Our experiments on the Product-10K dataset (5 queries) show:
>
> |Attack Method|AUC|TPR@1%FPR|TPR@10%FPR|
> |-|-|-|-|
> |No Attack|0.999|0.996|0.998|
> |Prompt Stripping|0.515|0.013|0.124|
>
> The results show that prompt stripping can effectively degrade our detection performance.
>
> **2. Practical Challenges**
>
> While prompt stripping significantly impacts detection performance, implementing such filtering in RAIG systems faces serious practical challenges:
> - Many legitimate use cases require precise alphanumeric inputs (product codes, brand names).
> - Our analysis on Product-10K identified numerous legitimate brand names that would be incorrectly filtered, such as "Royalstar", "YASEE", "IPASON", "LIEYANZHANDUI", "kocotree", "Aocilenda", and "NUTRICIA".
> - Such filtering would significantly degrade RAIG system utility in e-commerce and technical applications.
>
> We will include a detailed discussion of these limitations in our revised paper.
>
> >### **[How the method might handle W3.(a) and W3.(b)]**
>
> We propose two potential strategies to mitigate the vulnerability to OCR-based filtering and prompt stripping:
>
> **1. Using Meaningful Characters as Keys.**
> Instead of random character sequences, we could use meaningful words or phrases as sentinel keys. However, this approach faces two challenges:
> - Finding the right balance between protection effectiveness and normal user experience, as legitimate users might naturally use these meaningful keys in their queries
> - The meaningful keys might retrieve images related to their semantic meaning rather than images containing the actual text, potentially reducing detection accuracy
>
> **2. Embedding Keys in Natural Contexts.**
> We could embed random character keys in natural-looking contexts, such as "VasWiW brand cup" or "VasWiW restaurant". This approach could help bypass prompt filtering as these queries appear more legitimate.
>
> We plan to explore these mitigation strategies in our future work.
>
> >### **[Q2] Unintended Character Generation in Normal Usage**
>
> In Section 5.2 "Generation quality preservation", we investigated whether using protected images would affect normal generation capability. Among all queries, 8.3% retrieved sentinel images. We analyzed all generated images from these queries and found:
>
> 1. 36% of these images generated by SDXL-based RAIG after retrieving sentinel images did contain text characters.
>
> 2. These characters were not direct copies of the retrieval keys embedded in the retrieved sentinel images. Instead, they appeared with significant modifications and transformations, making the original characters unrecognizable.
>
> 3. The characters were naturally integrated into the generated scenes, appearing as store signs, ship decorations, or wall graffiti. This integration did not noticeably impact the visual quality or coherence of the generated images.
>
> In this test, we intentionally increased the number of sentinel images to 300 to evaluate the generation capability. In practice, only a small number of sentinel images (1-20 images in a reference dataset of 10,000 images) are needed for protection, which would reduce the likelihood of retrieving sentinel images during normal generation.
>
> We will include these visual examples and a discussion in our revised paper.

---

> > ### Comment · Reviewer_qQTS · 2025-08-04
> >
> > I thank the authors for their rebuttal which has addressed most of my interrogations. I particularly appreciated the effort of implementing the attacks (which were asked in many reviews) and the honesty of the answers.
> >
> > With the addition of this part on adversarial attacks (in the main paper or in the appendix) and with the additional experimental details, I strongly believe that this paper has its place at the conference and in the field. It could serve as a first baseline for protection of image datasets in the context of RAIG and opens up interesting problems to solve.

---

> > > ### Author Response · Authors · 2025-08-04
> > > **Thanks for your valuable feedback**
> > >
> > > Dear Reviewer qQTS,
> > >
> > > We sincerely appreciate your valuable feedback on our work, which makes this work more comprehensive. We're particularly grateful for recognizing our efforts in implementing the adaptive attacks. Your positive assessment of our paper's contribution to the field is highly encouraging. We will ensure all the discussed additions are properly incorporated in our paper.
> > >
> > > Best regards,
> > >
> > > The Authors

---

### Official Review · Reviewer_LHUd · 2025-07-03

**Clarity:** 3
**Significance:** 2
**Originality:** 3
**Rating:** 4
**Confidence:** 3

**Summary:**

This paper proposes a new approach to detect the unauthorized use of private image datasets in RAIG systems. The method involves augmenting a private dataset with synthesized "sentinel" images that are visually consistent with the dataset but also contain a unique, embedded textual key. A dataset owner can then probe a suspect RAIG system by providing these keys as prompts. High visual similarity between the generated output and the original sentinel image is used as evidence of dataset misuse. The work demonstrates high detection rates across several modern RAIG systems with minimal impact on authorized generation quality.

**Questions:**

- Can you discuss the robustness of ImageSentinel to an adaptive adversary who actively filters the dataset for suspicious images (e.g., via OCR)?
- Can you propose an experiment to decisively prove that the high similarity scores you observe are a result of retrieval and not just the direct, non-retrieval-based rendering of the key by the generator?
- How do you propose this method remains reliable against adversaries using models where this capability is weak or absent?

**Ethical Concerns:**

["NO or VERY MINOR ethics concerns only"]

**Final Justification:**

After productive discussions with the authors, all my concerns have been addressed. The new evaluation results provide more insights into the pros and cons of the proposed protection framework. Therefore, I choose to raise my score.

**Limitations:**

The authors include a "Limitations" section, which is appreciated. However, it omits several critical limitations raised in the weaknesses section, such as the vulnerability to adaptive adversaries.

**Paper Formatting Concerns:**

No major formatting issues were identified.

**Quality:**

3

**Strengths And Weaknesses:**

### Strengths
- This paper studies a pressing problem when a retrieval-augmented pipeline is commonly adopted by GenAI.
- The experimental setup and ablation studies are comprehensive, and the writing is clear.

### Weaknesses

- The evaluation lacks a defense against an adaptive adversary, a standard requirement for security claims. The entire defense hinges on the stealthiness of sentinels, yet it assumes a passive adversary. A motivated attacker aware of this strategy could easily employ countermeasures, such as using OCR to filter out any images containing conspicuous, random text strings before ingestion. The necessity of evaluating against such adaptive attacks is a well-established principle in the security community (e.g., [Tramèr et al., 2020], [Athalye et al., 2018]). Without this, the claimed robustness is unsubstantiated.

- Some baselines are not competitive, comprising known-to-fail "strawmen" while ignoring the state-of-the-art. The paper compares its method against traditional image watermarking (HiDDeN, FIN). However, if I understand it correctly, recent work [37] has already underscored that such watermarks are not robust to the complex transformations of modern generative models. These baseline watermark methods resulting in near-random performance (AUC $\approx$ 0.5) are therefore expected and do not constitute a meaningful benchmark. In my opinion, the paper avoids comparison with more relevant and challenging baselines like the membership attacks for RAG that the paper cites in its own related work (e.g., [25]).

- The evaluation of the detection mechanism is a bit ambiguous. The core claim is that the key triggers the retrieval of the sentinel. However, the evaluation fails to disentangle this from a more trivial explanation: that the RAIG's powerful generator (e.g., GPT-4o) simply renders the key from the prompt, leading to an image that is naturally similar to the sentinel. The "triggering rate" in Table 2, which is key to this claim, is not clearly defined for a black-box setting, and this ambiguity undermines the interpretation of the core results.

- My final primary concern is about the assumption of the proposed defense. Based on my understanding, the defense only works if the adversary's generator is powerful enough to faithfully render the key; if a weaker generator is used, the detection mechanism will likely fail.

### References
- [Tramèr et al., 2020] Florian Tramèr et al. On Adaptive Attacks to Adversarial Example Defenses. Proc. of NeurIPS, 2020.
- [Athalye et al., 2018] Anish Athalye et al. Obfuscated Gradients Give a False Sense of Security: Circumventing Defenses to Adversarial Examples. Proc. of ICML, 2018.

---

> ### Author Rebuttal · Authors · 2025-07-29
>
> We thank Reviewer LHUd for the rigorous analysis and valuable feedback on security concerns.
>
> >### **[W1, Q1] Defense against adaptive adversary: No evaluation against attackers who could use OCR to filter out images with random text strings.**
>
> We have conducted experiments to evaluate potential attacks that attempt to detect and filter sentinel images.
>
> **1. Attack Implementation**
>
> We implemented two attack strategies using OCR-based random character detection:
> - Filtering: removing detected sentinel images
> - Masking: obscuring detected text regions
>
> **2. OCR Detection Performance**
>
> We use three widely-used OCR systems (EasyOCR, Tesseract OCR, and PaddleOCR) on two datasets, and apply the PyEnchant library [1] to determine whether the detected text contains random character sequences. The results are shown below.
>
> |OCR System|Dataset|Sentinel Images Detection Rate|Clean Images Filtered Rate|
> |-|-|-|-|
> |EasyOCR|Product-10K|92.7%|41.7%|
> |EasyOCR|LLaVA dataset|90.2%|35.3%|
> |Tesseract|Product-10K|63.6%|39.7%|
> |Tesseract|LLaVA dataset|61.5%|23.7%|
> |PaddleOCR|Product-10K|98.3%|51.1%|
> |PaddleOCR|LLaVA dataset|97.5%|38.0%|
>
> While OCR-based attacks are possible to filter most sentinel images, the **high false positive rates** on clean images indicate that such attacks would significantly impair RAIG's normal functionality.
>
> [1] Ryan, K. PyEnchant: a spellchecking library for Python.
>
> **3. Impact Analysis**
>
> Detection performance under EasyOCR-based attacks (LLaVA Visual Instruct dataset):
>
> |Queries|Attack Method|AUC|TPR@1%FPR|TPR@10%FPR|
> |-|-|-|-|-|
> |5|No Attack|0.99|0.98|0.99|
> ||Filtering|0.63|0.16|0.34|
> ||Masking|0.62|0.15|0.34|
> |10|No Attack|1.0|1.0|1.0|
> ||Filtering|0.71|0.19|0.40|
> ||Masking|0.69|0.20|0.42|
> |50|No Attack|1.0|1.0|1.0|
> ||Filtering|0.90|0.67|0.80|
> ||Masking|0.91|0.65|0.82|
> |100|No Attack|1.0|1.0|1.0|
> ||Filtering|0.97|0.93|0.96|
> ||Masking|0.98|0.94|0.96|
>
> Generation quality impact (300 samples):
>
> |Attack Method|CLIP|SigLIP|DINO|
> |-|-|-|-|
> |No Attack|0.772|0.743|0.605|
> |Filtering|0.531|0.527|0.503|
> |Masking|0.559|0.546|0.511|
>
> These results demonstrate that while attacks can initially degrade detection effectiveness, this can be mitigated by increasing query numbers. Such attacks also compromise RAIG's generation quality, making them impractical for real-world deployment.
>
> We will discuss these potential adversaries in the limitations section, along with experimental results. Future work could explore more attack and defense strategies to further validate the robustness of our method.
>
> >### **[W2] Some baselines are not competitive. Comparison with more relevant and challenging baselines like membership attacks for RAG cited in related work.**
>
> Thank you for raising this important point about baselines.
>
> **1. Baseline Selection**
>
> Our work addresses dataset protection for Retrieval-Augmented Image Generation (RAIG) systems, where protection methods are still largely unexplored. Therefore, we explored **adapting** text RAG protection method (Ward [1]) to RAIG by creating baselines Ward-HiDDeN and Ward-FIN in our paper. We elaborate below on why other existing methods cannot be directly adapted to RAIG.
>
> **2. Challenges in Adapting RAG Methods to RAIG**
>
> We investigated the feasibility of converting existing RAG-related methods:
> - MIA for RAG [2] ([25] in main paper): This method relies on text-based LLMs to answer "Yes" or "No" for membership inference. However, in RAIG, the image generator replaces the text-based LLM and can only generate images rather than provide binary answers.
>
> - Ward [1] and [3]: These methods watermarks texts in the reference dataset and relies on the persistence of text watermarks in generated outputs for detection. Since our reference data consists of images, we adapted the text watermarking approach to image watermarking, creating baselines Ward-HiDDeN and Ward-FIN.
>
> - S^2MIA [4] and MBA [5]: These methods rely on using masked target documents as input and observing accuracy of RAG-generated content for the masked portions to determine if the target document exists in the reference dataset. When adapting to RAIG, we would need to input masked target images. However, this is infeasible since RAIG accepts text rather than image inputs.
>
> - Phantom [6]: This method depends on white-box attacks, which conflicts with our threat model's black-box assumptions.
>
> - TrojanRAG [7]: This method requires attackers to have the ability to train the RAG system. However, since we directly use existing black-box models, this approach cannot be effectively adapted.
>
> [1] Jovanović, N. et al. "Ward: Provable rag dataset inference via llm watermarks." ICLR 2025.
>
> [2] Anderson, M. et al. "Is My Data in Your Retrieval Database? Membership Inference Attacks Against Retrieval Augmented Generation." ICISSP 2025.
>
> [3] Liu, Y. et al. "Dataset protection via watermarked canaries in retrieval-augmented llms." arXiv 2025.
>
> [4] Li, Y. et al. "Generating is believing: Membership inference attacks against retrieval-augmented generation." ICASSP 2025.
>
> [5] Liu, M. et al. "Mask-based membership inference attacks for retrieval-augmented generation." WWW 2025.
>
> [6] Chaudhari, H. et al. "Phantom: General trigger attacks on retrieval augmented language generation." arXiv 2024.
>
> [7] Cheng, P. et al. "TrojanRAG: Retrieval-augmented generation can be backdoor driver in large language models." arXiv 2024.
>
> **3. Additional Baseline**
>
> To provide a more comprehensive evaluation, we implemented a semantic-based baseline that uses semantic descriptions as retrieval keys instead of character sequences. This baseline represents a more intuitive approach that leverages semantic descriptions to detect unauthorized dataset usage. Specifically, we:
>
> 1) Extract semantic descriptions from images.
> 2) Use these descriptions as retrieval keys to query the RAIG system.
> 3) Compare generated images with original images to detect unauthorized use.
>
> As shown in Table 2 of main paper, most semantic descriptions (more than 60%) as retrieval keys bypass retrieval as their generators create satisfactory images directly, preventing detection of target images in the reference dataset.
>
> We further evaluated this baseline on the LLaVA Visual Instruct Pretrain Dataset:
>
> |RAIG|Query|ImageSentinel|||Semantic-based|||
> |-|-|-|-|-|-|-|-|
> |||AUC|T@1%F|T@10%F|AUC|T@1%F|T@10%F|
> |SDXL|3|0.974|0.934|0.958|0.623|0.200|0.356|
> ||10|1.000|1.000|1.000|0.785|0.340|0.563|
> ||20|1.000|1.000|1.000|0.876|0.540|0.717|
> |OmniGen|3|0.873|0.584|0.744|0.634|0.209|0.327|
> ||10|0.989|0.922|0.974|0.786|0.373|0.567|
> ||20|1.000|0.996|1.000|0.896|0.572|0.763|
>
> The results demonstrate that our approach outperforms the semantic-based baseline. We will include these results in the main paper for a more comprehensive comparison.
>
>
> >### **[W3, Q2] (W3) Detection mechanism validation: Need to verify whether high similarity comes from retrieval or direct key rendering. (Q2) Validating Retrieval vs Direct Rendering.**
>
> We appreciate these important questions about our evaluation methodology.
>
> **1. Analysis of Detection Mechanism**
>
> Direct key rendering would be insufficient for detection.
>
> Our detection mechanism does not rely on rendering keys from prompts. Instead, we detect unauthorized use through measuring **visual similarity** between RAIG-generated images and our sentinel images. The design the key as random characters (e.g., "VasWiW") without any information about corresponding sentinel image's semantical attributes.
>
> Therefore, it is theoretically impossible for generative models to recreate the specific visual characteristics of sentinel images solely from these random characters. The high similarity scores can only be achieved through successful retrieval of the sentinel images.
>
>
> **2. **[Q2]** Experiments for Direct Key Rendering**
>
> We conducted a controlled experiment comparing generation with retrieval enabled versus disabled (direct generation) using the same keys. We use 10 queries for LLaVA Visual Instruct Pretrain Dataset:
>
> |Setting|AUC|T@1%F|T@10%F|
> |-|-|-|-|
> |Retrieval enabled|1.000|1.000|1.000|
> |Retrieval disabled|0.512|0.008|0.102|
>
> The results are expected since our keys are random character sequences with no semantic correlation to sentinel images' visual content, making it impossible for direct generation to reproduce specific visual elements without retrieval.
>
> **3. Definition of Triggering Rate**
>
> The 'triggering rate' is defined as the proportion of queries where a RAIG system initiates the retrieval process when given a key. This metric is important since some RAIG systems [1] may bypass retrieval if their generators believe they can directly generate satisfactory images. Our method ensures a high triggering rate by carefully designing the key.
>
> Triggering rate is an intermediate metric used during our experimental analysis. In actual deployment (black-box setting), we only need to measure the semantic similarity between generated images and our sentinel images to detect unauthorized use. As shown in Table 3, this detection achieves high performance with black-box setting.
>
> [1] Shalev-Arkushin, R. et al. "ImageRAG: Retrieval-Augmented Image Generation with Semantic Guidance." arXiv 2025.
>
> >### **[W4, Q3] Dependency of Defense Mechanism on Generator Capability.**
>
> We demonstrated that high similarity scores cannot be achieved through direct key rendering in response to **[W3, Q2]**.
>
> Our keys are random character sequences with no semantic correlation to visual content of sentinel images. Therefore, regardless of the generator's capability, it cannot directly render specific visual elements of sentinel images solely from these keys. The high similarity scores can only be achieved through the successful retrieval of sentinel images.
>
> The requirement for RAIG systems is that the generator can incorporate semantic features from retrieved images into its output, which is a fundamental capability that functional RAIG system possesses.

---

> > ### Comment · Reviewer_LHUd · 2025-08-03
> >
> > Thank you for the detailed responses, which have partially addressed my concerns.
> >
> > I still have questions regarding the robustness against adaptive attacks:
> >
> > 1) My primary concern is the design of the attack method. In my view, the PyEnchant library is unsuitable for this scenario. PyEnchant is designed to detect misspellings, not random strings. Inaccurate OCR results or typos in images could lead to false positives. A more appropriate approach would be to train a lightweight model to determine if input strings are randomly generated or to use an off-the-shelf LLM to directly assess whether OCR-detected strings are random.
> >
> > 2) Given the small proportion of sentinel images (as noted in the response to Reviewer qQTS), why does the filtering attack method cause such a significant reduction in generation quality? Does this indicate that the filters incorrectly block many images? It would be helpful if the authors could provide more details on the additional results, such as the number of false and true positives.

---

> > > ### Author Response · Authors · 2025-08-04
> > > **Our Response (1/2)**
> > >
> > > We appreciate the reviewer's valuable suggestion regarding attack method design. We have conducted additional experiments using GPT-4.1 mini for more accurate random string detection.
> > >
> > > **1. Detection Method**
> > >
> > > We use the following prompt for GPT-4.1 mini to determine if OCR-detected text contains random strings:
> > >
> > > ```
> > > prompt = (
> > >     "Please determine if the following text (extracted by OCR) contains random/meaningless characters or gibberish. Consider it random if:\n"
> > >     "1. It's not a common word, brand name, or recognizable abbreviation\n"
> > >     "2. It has no clear pattern or structure\n"
> > >     "3. It appears to be a random combination of letters\n\n"
> > >     "Note: Ignore minor OCR mistakes (like '0' for 'O', '1' for 'I', etc.) and common typos. Only focus on whether the underlying text appears meaningful or random.\n\n"
> > >     "Answer only with 'yes' if it's random, or 'no' if it has clear meaning or structure.\n\n"
> > >     f"Text: {text}"
> > > )
> > > ```
> > >
> > > **Detection Performance Using GPT-4.1 mini**
> > >
> > > |Queries|Attack Method|AUC|TPR@1%FPR|TPR@10%FPR|
> > > |-|-|-|-|-|
> > > |5|No Attack|0.99|0.98|0.99|
> > > ||Filtering attack (GPT-4.1 mini)|0.60|0.14|0.30|
> > > |10|No Attack|1.0|1.0|1.0|
> > > ||Filtering attack (GPT-4.1 mini)|0.66|0.18|0.37|
> > > |50|No Attack|1.0|1.0|1.0|
> > > ||Filtering attack (GPT-4.1 mini)|0.86|0.52|0.70|
> > > |100|No Attack|1.0|1.0|1.0|
> > > ||Filtering attack (GPT-4.1 mini)|0.96|0.88|0.92|
> > >
> > > Similar to our previous findings with PyEnchant-based filtering, the results show that while the GPT-4.1 mini-based filtering attack can degrade detection effectiveness, this impact can be mitigated by increasing the number of queries.

---

> > > > ### Author Response · Authors · 2025-08-04
> > > > **Our Response (2/2)**
> > > >
> > > > **2. Regarding the significant reduction in generation quality**
> > > >
> > > > While sentinel images only comprise very small portion of the dataset, the significant quality degradation is explained by the **high false positive rates** of the filtering methods, which result in incorrectly removing a large number of legitimate images. We presented False/True Positives (as "Sentinel Images Detection Rate"/"Clean Images Filtered Rate") in the response to [W1, Q1]. Below we provide a comprehensive analysis of false and true positive rate for both PyEnchant-based and GPT-4.1 mini-based filtering attacks:
> > > >
> > > > |OCR System|Dataset|Attack Method|True Positive Rate|False Positive Rate|
> > > > |-|-|-|-|-|
> > > > |EasyOCR|Product-10K|Filtering attack (PyEnchant)|92.7% (1,854/2,000)|41.7% (4,173/10,000)|
> > > > |||Filtering attack (GPT-4.1 mini)|92.5% (1,849/2,000)|32.8% (3,275/10,000)|
> > > > |EasyOCR|LLaVA dataset|Filtering attack (PyEnchant)|90.2% (1,803/2,000)|35.3% (3,531/10,000)|
> > > > |||Filtering attack (GPT-4.1 mini)|90.1% (1,801/2,000)|30.1% (3,011/10,000)|
> > > > |PaddleOCR|Product-10K|Filtering attack (PyEnchant)|98.3% (1,966/2,000)|51.1% (5,114/10,000)|
> > > > |||Filtering attack (GPT-4.1 mini)|98.1% (1,961/2,000) |33.1% (3,308/10,000)|
> > > > |PaddleOCR|LLaVA dataset|Filtering attack (PyEnchant)|97.5% (1,950/2,000)|38.0% (3,796/10,000)|
> > > > |||Filtering attack (GPT-4.1 mini)|97.4% (1,947/2,000) |20.3% (2,027/10,000)|
> > > >
> > > > As shown in the results, while both methods effectively detect sentinel images, they suffer from substantial false positive rates, with GPT-4.1 mini showing improved but still significant filtering of legitimate images.
> > > >
> > > > Generation quality impact with different filtering approaches:
> > > >
> > > > |Attack Method|CLIP|SigLIP|DINO|
> > > > |-|-|-|-|
> > > > |No Attack|0.772|0.743|0.605|
> > > > |Filtering attack (PyEnchant)|0.531|0.527|0.503|
> > > > |Filtering attack (GPT-4.1 mini)|0.683|0.662|0.557|
> > > >
> > > > The results confirm that high false positive rates significantly impact generation quality. While GPT-4.1 mini-based filtering shows less degradation compared to PyEnchant, both methods reduce the generation performance from the baseline.
> > > >
> > > > **Analysis of High False Positive Rate**
> > > >
> > > > The high false positive rate occurs because many real-world images contain text elements that can be misidentified as random strings. We study some incorrectly filtered images in LLaVA Visual Instruct Pretrain Dataset. Based on our preliminary analysis, we identify five main challenges in accurately detecting legitimate text across different OCR systems and filtering methods (including PyEnchant and GPT-4.1 mini):
> > > >
> > > > 1. Image Quality Issues: Many real-world images contain text that is difficult to extract due to low resolution (e.g., 000000316.jpg, 000000537.jpg, 000000901.jpg), viewing angles (e.g., 000000520.jpg), or partial occlusion (e.g., 000000302.jpg, 000000415.jpg, 000001260.jpg).
> > > >
> > > > 2. Stylized Typography: Artistic or decorative fonts often lead to inaccurate text extraction and misclassification (e.g., 000000040.jpg, 000000141.jpg, 000000476.jpg, 000000772.jpg).
> > > >
> > > > 3. Non-semantic Watermarks: Some image watermarks and identifiers inherently lack semantic meaning but serve valid identification purposes (e.g., 000000341.jpg, 000001326.jpg).
> > > >
> > > > 4. Brand Name Recognition: Certain brand names in English may not be recognized as meaningful words by dictionary-based approaches (e.g., 000000946.jpg).
> > > >
> > > > 5. Complex Backgrounds: Watermarks that blend with image backgrounds pose significant challenges for accurate text extraction (e.g., 000001207.jpg).
> > > >
> > > > We also observe several scenarios where GPT-4.1 mini demonstrates improvements over the PyEnchant-based approach:
> > > >
> > > > 1. Incomplete OCR Results: Enhanced recognition of partially captured text fragments (e.g., 000000047.jpg, 000000115.jpg, 000000199.jpg, 000000378.jpg) that conventional filters might flag as random strings.
> > > >
> > > > 2. Brand Names and Person Names: More accurate identification of nouns (e.g., 000000012.jpg, 000000224.jpg) that may not appear in dictionaries.
> > > >
> > > > 3. Compound Words and Closely Spaced Text: Better handling of compound nouns, especially in website addresses, particularly common in website addresses (e.g., 000000073.jpg, 000000313.jpg, 000000534.jpg) and cases where OCR merges multiple words due to proximity (e.g., 000000239.jpg), which GPT-4.1 mini can correctly recognize.
> > > >
> > > > We will provide a more comprehensive analysis in the final version of our paper. We plan to incorporate these attack methods into benchmark for this field, and release them in our repository.

---

> > > > > ### Comment · Reviewer_LHUd · 2025-08-04
> > > > >
> > > > > Thank you so much for the comprehensive follow-ups! Everything is clear to me, except that I am still a bit confused about why the number of sentinel images was increased to 2,000 for evaluating TPRs in the adaptive attack. This is significantly higher than the number of sentinel images ($\leq$ 30) reported in the original paper.
> > > > >
> > > > > However, my above question is a minor issue, and I will raise my score. I hope the authors can include discussions on potential adaptive attacks in the next version, as highlighting the limitations and risks of protection failures is valuable for the audience. This does not necessarily imply weak protections.
> > > > >
> > > > > Thank you again.

---

> > > > > > ### Author Response · Authors · 2025-08-04
> > > > > > **Thank you for your helpful feedback**
> > > > > >
> > > > > > Thank you for your feedback and for agreeing to raise the score.
> > > > > >
> > > > > > The 2,000 sentinel images are solely used for computing reliable statistics on true positive rates when evaluating the sentinel image detection performance of adaptive attack methods. For computing dataset misuse detection metrics of our protection method (AUC, TPR@1%FPR, TPR@10%FPR), we always use a small number of sentinel images, including those involving adaptive attack evaluations. For example, in our last response, we use 5-100 sentinel images for misuse detection evaluation, as shown in the Detection Performance table. We will clarify these implementation details more explicitly in our paper.
> > > > > >
> > > > > > We appreciate your suggestion about the important adaptive attacks, and we will include detailed discussions on the limitations and risks in the next version.
> > > > > >
> > > > > > Thank you again for helping us improve the completeness and clarity of our paper.

---

> > > > > > > ### Comment · Reviewer_LHUd · 2025-08-06
> > > > > > >
> > > > > > > Thank you for your active engagement in the discussion. I learned a lot during this process. Good luck!

---

### Official Review · Reviewer_GDVj · 2025-07-05

**Clarity:** 3
**Significance:** 3
**Originality:** 3
**Rating:** 5
**Confidence:** 1

**Summary:**

The paper proposes a framework (ImageSentinel) to protect image datasets from unauthorised use in RAIG systems. It inserts sentinel images containing random character sequences as triggers for detecting unauthorised dataset use. The authors demonstrate their framework by showing reliable detection of dataset misuse across multiple RAIG systems in a black-box setting.

**Questions:**

* Have you tested whether an adversary with access to the protected dataset could filter or detect the sentinel images before integrating them into a RAIG system?
* How would your approach perform if the RAIG system filtered or sanitized the random character prompts before retrieval (e.g., by stripping non-semantic input)? How much would the detection rate drop when such preprocessing is applied to the prompts?
* Can you clarify how many queries and how much compute/time is required for reliable detection? What are the limits of detection with fewer queries?

**Ethical Concerns:**

["NO or VERY MINOR ethics concerns only"]

**Final Justification:**

As stated in the discussion, I will keep my score as is.

**Limitations:**

As described above

**Quality:**

3

**Strengths And Weaknesses:**

The paper addresses an important and timely challenge, namely the unathorised use of image (datasets), specifically with RAIG systems. Here, existing watermarking approaches fall short, making this an important contribution to the topic of RAIG image retrieval and copyright in AI. I appreciate that the authors show how their approach can work without disrupting overall RAIG for general usage. The experiments and evaluation appear quite comprehensive, they test across multiple RAIG systems and relevant ablations.

There are a limitations with regards to the real-world testing of the models, namely evaluations are performed on synthetic datasets, the threat from proprietary or fine-tuned RAIG systems in the wild is not assessed. In a similar vein, but rather as a note than a criticism of the method overall, the approach here relies on black-box querying, which requires repeated queries to the suspected RAIG system, which may not be feasible in a practical setting.

Further, the authors could extend their evaluation of the methodology by providing adverserial analysis, specifically w.r.t. the detectability of sentinel images; malicious users could potentially filter or recognise sentinel patterns.

---

> ### Author Rebuttal · Authors · 2025-07-29
>
> We sincerely thank Reviewer GDVj for the thorough review and constructive suggestions.
>
> >###  **[W1] The evaluations are only performed on synthetic datasets; there is no assessment of threats from proprietary or fine-tuned RAIG systems in real-world scenarios.**
>
> **For datasets:**
>
> Our evaluation covers both synthetic and real-world datasets:
>
> 1. Our main experiments used the LLaVA Visual Instruct Dataset [1], which contains real-world images from diverse sources and scenarios, including natural scenes, everyday objects, people, and more.
> 2. We also evaluated ImageSentinel on Product-10K [2] in the supplementary materials, a real-world commercial dataset with 30,000 product images including both in-shop photos and customer-generated images.
>
> We will move more Product-10K experiments to the main paper to strengthen our evaluation by including more real-world e-commerce applications.
>
> [1] Liu, Haotian, et al. "Visual instruction tuning." NeurIPS 2023.
>
> [2] Bai, Yalong, et al. "Products-10k: A large-scale product recognition dataset." arXiv preprint 2020.
>
> **For RAIG systems:**
>
> 1. We evaluated ImageSentinel on both proprietary and open-source RAIG systems: Proprietary system (GPT-4o) and Open-source systems (SDXL and OmniGen).
>
> 2. Following the common practice in retrieval-augmented generation research [3, 4] and widely-used frameworks like LangChain [5], we use pre-trained generators without fine-tuning. As RAIG is an emerging field, fine-tuned RAIG systems are not yet widely available in the community. We will extend our evaluation to fine-tuned variants when they become available.
>
> [3] Shalev-Arkushin, Rotem, et al. "ImageRAG: Retrieval-Augmented Image Generation with Semantic Guidance." arXiv preprint 2025.
>
> [4] Ru, Dongyu, et al. "RagChecker: A fine-grained framework for diagnosing retrieval-augmented generation." NeurIPS 2024.
>
> [5] Mavroudis, Vasilios. "LangChain v0.3." preprint 2024.
>
>
> >### **[W2] The approach relies on black-box querying, requiring repeated queries to the suspected RAIG system, which may not be feasible in practical settings.**
>
> We appreciate this practical concern. We discuss the feasibility of our black-box query approach as follows:
>
> **1. Minimal Query Requirements**
>
> As shown in Table 3, S3 and S4 in the main paper and supplementary materials, our approach achieves efficient detection with very few queries. For example, only 5 queries are needed to reach 99.6% TPR at 1% FPR on Products-10k (total 30,000 reference images).
>
> **2. Established Practicality**
>
> Query-based approaches have been widely adopted in retrieval-augmented systems for backdoor attack detection [6], membership inference [7], and dataset protection [8]. Such querying overhead occurs only during investigation of suspected misuse, not as a continuous requirement.
>
> [6] Chen, Zhaorun, Zhen Xiang, Chaowei Xiao, Dawn Song, and Bo Li. "AgentPoison: Red-teaming LLM agents via poisoning memory or knowledge bases." NeurIPS 2024.
>
> [7] Liu, Mingrui, Sixiao Zhang, and Cheng Long. "Mask-based membership inference attacks for retrieval-augmented generation." WWW 2025.
>
> [8] Liu, Yepeng, Xuandong Zhao, Dawn Song, and Yuheng Bu. "Dataset Protection via Watermarked Canaries in Retrieval-Augmented LLMs." ICML 2025 Workshop R2-FM.
>
> >### **[W3, Q1] The authors should further extend their evaluation with adversarial analysis, specifically regarding the detectability of sentinel images; malicious users could potentially filter or recognize sentinel patterns.**
>
> We appreciate the reviewer's suggestion about adversarial analysis. We have conducted experiments to evaluate potential attacks that attempt to detect and filter sentinel images.
>
> **1. Attack Implementation**
>
> We implemented two attack strategies using OCR-based random character detection:
> - Filtering: removing detected sentinel images
> - Masking: obscuring detected text regions
>
> **2. OCR Detection Performance**
>
> We use three widely-used OCR systems (EasyOCR, Tesseract OCR, and PaddleOCR) on two datasets, and apply the PyEnchant library [1] to determine whether the detected text contains random character sequences. The results are shown below.
>
> |OCR System|Dataset|Sentinel Images Detection Rate|Clean Images Affected Rate|
> |------------|---------|----------------|----------------------|
> |EasyOCR|Product-10K|92.7%|41.7%|
> |EasyOCR|LLaVA Visual Instruct|90.2%|35.3%|
> |Tesseract OCR|Product-10K|63.6%|39.7%|
> |Tesseract OCR|LLaVA Visual Instruct|61.5%|23.7%|
> |PaddleOCR|Product-10K|98.3%|51.1%|
> |PaddleOCR|LLaVA Visual Instruct|97.5%|38.0%|
>
> While OCR-based attacks are possible to filter most sentinel images, the **high false positive rates** on clean images indicate that such attacks would significantly impair RAIG's normal functionality. This is particularly problematic for datasets like Product-10K, where a substantial portion of legitimate images containing product names, URLs, or watermarks would be incorrectly filtered out.
>
> [1] Ryan, K. PyEnchant: a spellchecking library for Python.
>
> **3. Impact Analysis**
>
> Detection performance under EasyOCR-based attacks (LLaVA Visual Instruct dataset):
>
> |Queries|Attack Method|AUC|TPR@1%FPR|TPR@10%FPR|
> |---------|--------------|------|------------|------------|
> |5|No Attack   |0.99|0.98      |0.99      |
> ||Filtering   |0.63|0.16      |0.34      |
> ||Masking     |0.62|0.15      |0.34      |
> |10|No Attack   |1.0 |1.0       |1.0       |
> ||Filtering   |0.71|0.19      |0.40      |
> |      |Masking     |0.69|0.20      |0.42      |
> |50     |No Attack   |1.0 |1.0       |1.0       |
> |       |Filtering   |0.90|0.67      |0.80      |
> |       |Masking     |0.91|0.65      |0.82      |
> |100    |No Attack   |1.0 |1.0       |1.0       |
> |       |Filtering   |0.97|0.93      |0.96      |
> |       |Masking     |0.98|0.94      |0.96      |
>
> Generation quality impact (300 samples):
>
> |Attack Method|CLIP|SigLIP|DINO|
> |--------------|-------|----------|--------|
> |No Attack   |0.772|0.743   |0.605 |
> |Filtering   |0.531|0.527   |0.503 |
> |Masking     |0.559|0.546   |0.511 |
>
> - These results demonstrate that while attacks can initially degrade detection effectiveness, this can be mitigated by increasing query numbers. For example, under filtering attacks, the AUC drops to 0.63 with 5 queries, but recovers to 0.97 with 100 queries.
> - Such attacks significantly compromise RAIG's generation quality, making them impractical for real-world deployment. Under attacks, the CLIP scores drop from 0.772 to around 0.531 (Filtering attack) and 0.559 (Masking attack), while SigLIP scores decrease from 0.743 to around 0.53 (Filtering attack) and 0.55 (Masking attack), representing a degradation in generation quality.
>
> We will discuss these adversarial analyses in the limitations section, along with detailed experimental results. Future work could explore more sophisticated attack strategies and corresponding defense mechanisms to further validate the robustness of our protection framework.
>
> >### **[Q2] How would your approach perform if the RAIG system filtered or sanitized the random character prompts before retrieval (e.g., by stripping non-semantic input)? How much would the detection rate drop when such preprocessing is applied to the prompts?**
>
> We have conducted additional experiments to evaluate our approach's robustness against prompt preprocessing:
>
> **1. Prompt Stripping Attack**
>
> We implemented a **prompt stripping** attack that removes random character sequences in the queries.
>
> Our experiments on the Product-10K dataset (5 queries) show:
>
> |Attack Method|AUC|TPR@1%FPR|TPR@10%FPR|
> |-|-|-|-|
> |No Attack|0.999|0.996|0.998|
> |Prompt Stripping|0.515|0.013|0.124|
>
> The results show that prompt stripping can effectively degrade our detection performance.
>
> **2. Practical Challenges**
>
> While prompt stripping significantly impacts detection performance, implementing such filtering in RAIG systems faces serious practical challenges:
> - Many legitimate use cases require precise alphanumeric inputs (product codes, brand names).
> - Our analysis on Product-10K identified numerous legitimate brand names that would be incorrectly filtered, such as "Royalstar", "YASEE", "IPASON", "LIEYANZHANDUI", "kocotree", "Aocilenda", and "NUTRICIA".
> - Such filtering would significantly degrade RAIG system utility in e-commerce and technical applications.
>
> We will include a detailed discussion of these limitations and potential mitigation strategies in our revised paper.
>
> >### **[Q3] Can you clarify how many queries and how much compute/time is required for reliable detection? What are the limits of detection with fewer queries?**
>
> We thank the reviewer for this important question about the practical aspects of our detection method.
>
> **1. Query Requirements for Reliable Detection**
>
> As shown in Table 3, S3, and S4, ImageSentinel achieves reliable detection with very few queries. We list some results in Table S4 as:
>
> |Query Number|AUC|TPR@1%FPR|TPR@10%FPR|
> |--------------|-----|------------|-------------|
> |1|0.870|0.704|0.754|
> |3|0.989|0.944|0.982|
> |5|0.999|0.996|0.998|
>
> To address the limits with fewer queries, we specifically tested the performance with just one query, which still achieves a reasonable AUC of 0.870 and TPR of 0.754 at 10% FPR, though with reduced reliability compared to using more queries.
>
> **2. Computational Cost**
>
> The main computational overhead comes from image generation time per query:
>
> |RAIG System|Per-query Time|
> |-------------|---------------|
> |SDXL|54.5s|
> |OmniGen|156.2s|
> |GPT-4o|57.7s|
>
> With parallel processing (common in real-world deployments), total detection time approximates one query duration. Without parallelization, total time equals per-query time multiplied by query number.
>
> Given that reliable detection requires only 3-10 queries and supports parallel processing, the computational cost remains practical for practical applications.

---

> > ### Comment · Reviewer_GDVj · 2025-08-04
> >
> > I would like to thank the authors for their response. I believe my rating adequately reflects my evaluation, especially given the additional details provided by the authors.

---

> > > ### Author Response · Authors · 2025-08-04
> > > **Thanks for your valuable feedback**
> > >
> > > Dear Reviewer GDVj,
> > >
> > > We sincerely appreciate your time and careful evaluation of our work. We're grateful for your positive rating and will incorporate your suggestions in our paper.
> > >
> > >
> > > Best regards,
> > >
> > > The Authors

---

### Note · Authors · 2025-08-12

Dear Reviewers, ACs, and SACs,

We sincerely thank all reviewers, ACs, and SACs for your time and dedication in reviewing our paper. We are particularly encouraged that **all reviewers have indicated positive ratings following our discussions**, with **Reviewer LHUd** and **Reviewer yaqi** specifically raising their scores after our responses.

During the rebuttal period, we addressed the raised concerns and clarified our methodology and evaluation in line with the points reviewers raised. In particular, we focused on adaptive adversaries. We implemented and analyzed realistic attack strategies (text detection/filtering, prompt sanitization, similarity-based deduplication, and detect-and-inpaint), evaluated their impact and practicality, discussed mitigations, and expanded the limitations accordingly. These additions have significantly strengthened our paper by providing a more comprehensive analysis of our method's robustness while maintaining transparency about potential vulnerabilities.

We deeply appreciate the active engagement of all reviewers and will incorporate all the suggested improvements in our final version, including expanded experimental results and a thorough discussion of potential attack strategies.

Thank you again for helping us strengthen this work.



Best regards,

The Authors

---

### Decision · Program_Chairs · 2025-09-17

**Decision:**

Accept (poster)

**Comment:**

The paper has received unanimous final recommendations to accept, two on the borderline and two firm.  The borderline reviewers have raised their scores during the discussion phase.

The majority of reviewers see ImageSentinels as a timely contribution given the rising interest in RAIG systems though all had questions over practical attacks and deployment.  There were engagements post-rebuttal around adaptive adversarial attacks and use of OCR that have strengthened the rigor and practical applicability of the work and should be included in the final version.  Overall the AC agrees with the reviewer consensus to accept the paper - congratulations.